# Unsupervised Discovery of Object-Centric Neural Fields

**Rundong Luo**[*][†]                                                     *rundongluo@cs.cornell.edu*
*Cornell University*

**Hong-Xing Yu**[*]                                                          *koven@cs.stanford.edu*
*Stanford University*

**Jiajun Wu**                                                                *jiajunwu@cs.stanford.edu*
*Stanford University*

**Reviewed on OpenReview:** *https://openreview.net/forum?id=ScEv13W2f1*

## Abstract

We study inferring 3D object-centric scene representations from a single image. While recent methods have shown potential in unsupervised 3D object discovery, they are limited in generalizing to unseen spatial configurations. This limitation stems from the lack of translation invariance in their 3D object representations. Previous 3D object discovery methods entangle objects' intrinsic attributes like shape and appearance with their 3D locations. This entanglement hinders learning generalizable 3D object representations. To tackle this bottleneck, we propose the unsupervised discovery of Object-Centric neural Fields (uOCF), which integrates translation invariance into the object representation. To allow learning object-centric representations from limited real-world images, we further introduce a learning method that transfers object-centric prior knowledge from a synthetic dataset. To evaluate our approach, we collect four new datasets, including two real kitchen environments. Extensive experiments show that our approach significantly improves generalization and sample efficiency, and enables unsupervised 3D object discovery in real scenes. Notably, uOCF demonstrates zero-shot generalization to unseen objects from a single real image. The project page is available at https://red-fairy.github.io/uOCF/.

## 1 Introduction

Creating factorized, object-centric 3D scene representations is a fundamental ability in human vision and a long-standing topic of interest in computer vision and machine learning. Some recent work has explored unsupervised learning of 3D factorized scene representations from images alone (Stelzner et al., 2021; Yu et al., 2022; Smith et al., 2023; Jia et al., 2023). These methods have delivered promising results in 3D object discovery and reconstruction from a simple synthetic image.

However, existing methods fail to generalize to unseen spatial configurations and objects. A fundamental bottleneck is that their representations lack the invariance to the 3D positions of the objects. In particular, existing methods represent 3D objects as implicit functions in the viewer's coordinate frame, so that any change related the coordinate frame (e.g., slight changes in an object's location or subtle camera movements) may lead to significant changes in the object representation even if the object remains the same. Therefore, existing methods do not generalize when an object appears at an unseen location during inference.

To address this fundamental bottleneck, we propose the unsupervised discovery of Object-Centric neural Fields (uOCF). Unlike existing methods, uOCF explicitly infers an object's 3D location, disentangling it from the object's latent representation. This design builds translation invariance into the object representation, so

---

[*]Equal Contribution.
[†]Work done while R. Luo was a visiting student at Stanford University.

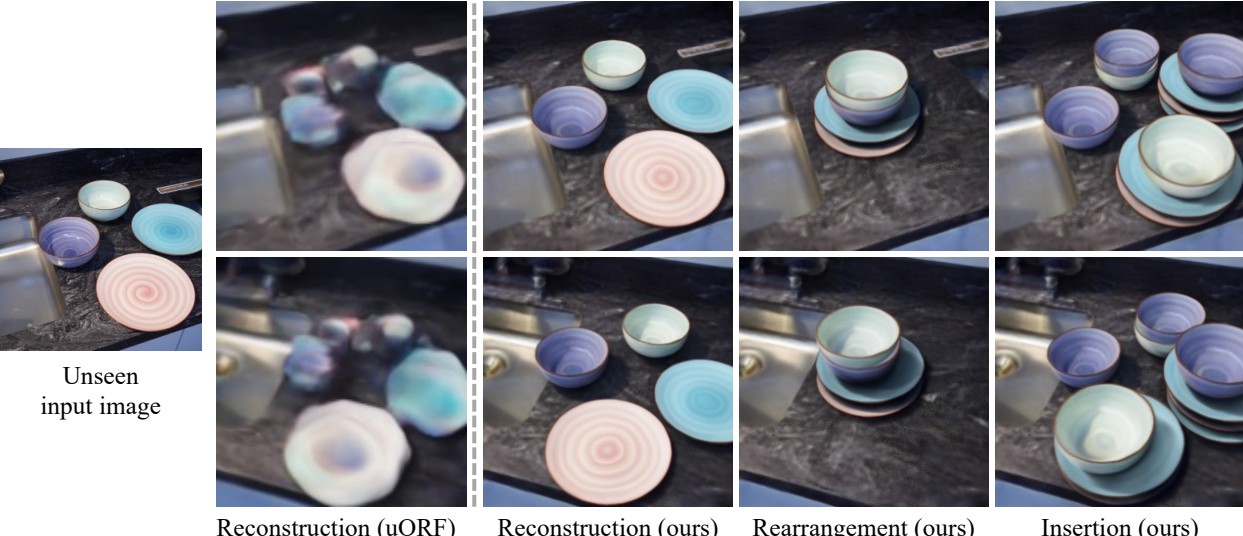

Figure 1: We propose the unsupervised discovery of Object-Centric neural Fields (uOCF), which infers factorized 3D scene representations from an unseen real image, thus enabling scene reconstruction and manipulation from novel views. We compare uOCF with the state-of-the-art method, uORF (Yu et al., 2022).

that the object's latent only represents the intrinsics of the object (e.g., shape and appearance). This design significantly improves generalization. As showcased in Figure 1, uOCF can generalize to unseen real-world scenes. We train uOCF on sparse multi-view images without object annotations. During inference, uOCF takes in a single image and generates a set of object-centric neural radiance fields (NeRFs) (Mildenhall et al., 2020) and a background NeRF.

Another advantage of our translation-invariant 3D object representation is that it facilitates learning 3D object priors from simple scenes and generalizes to more complex scenes with unseen spatial configurations and objects. This further boosts sample efficiency and thus it is particularly beneficial when we deal with real scenes where training data is often limited. We introduce an object prior learning method to this end.

To evaluate our approach, we introduce new challenging datasets for 3D object discovery, including two real kitchen datasets and two synthetic room datasets. The two real datasets feature real-world kitchen backgrounds and objects from multiple categories. The synthetic room datasets feature furniture with diverse, realistic shapes and textures. Across all these datasets, uOCF yields high-fidelity discovery of object-centric neural fields, allowing applications such as unsupervised 3D object segmentation and scene manipulation from a real image. uOCF shows strong generalization to unseen spatial configurations and high sample efficiency, and we showcase that it even allows *zero-shot* 3D object discovery on a few simple real scenes with unseen objects. In summary, our contributions are threefold:

- First, we highlight the overlooked role of translation invariance in unsupervised 3D object discovery. We instantiate the idea by proposing the unsupervised discovery of Object-Centric neural Fields (uOCF), which builds translation invariance to the object representation.

- Second, we introduce a 3D object prior learning method, which leverages uOCF's translation-invariant property to learn category-agnostic object priors from simple scenes and generalize to different object categories and scene layouts.

- Lastly, we collect four challenging datasets, Room-Texture, Room-Furniture, Kitchen-Matte, and Kitchen-Shiny, and show that uOCF significantly outperforms existing methods on these datasets, unlocking zero-shot, single-image object discovery. All code and data will be made public.

## 2 Related Works

**Unsupervised 2D object discovery.** Before the advent of deep learning, traditional methods for object discovery (often referred to as co-segmentation) primarily focused on locating visually similar objects across

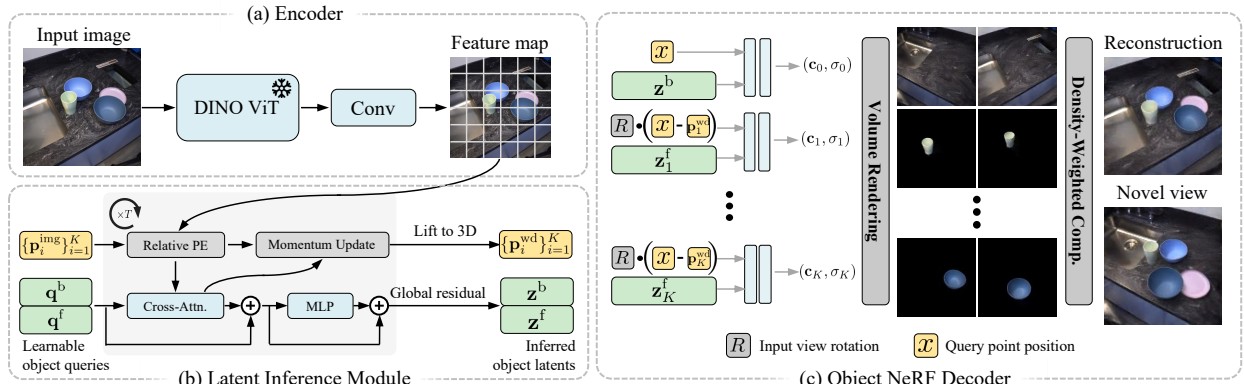

Figure 2: With a single forward pass, uOCF processes a single image input to infer a set of object-centric radiance fields along with their 3D locations and background radiance field. uOCF is trained on sparse multi-view images from a collection of scenes and uses a single image as input during inference.

a collection of images (Sivic et al., 2005; Russell et al., 2006), where objects are defined as visual words or clusters of patches (Grauman & Darrell, 2006; Joulin et al., 2010). This clustering concept was later incorporated into deep learning techniques for improved grouping results (Li et al., 2019; Vo et al., 2020). The incorporation of deep probabilistic inference propelled the field towards factorized scene representation learning (Eslami et al., 2016). These methods decompose a visual scene into several components, where objects are often modeled as latent codes that can be decoded into image patches (Kosiorek et al., 2018; Crawford & Pineau, 2019; Jiang et al., 2020; Lin et al., 2020), phase values (Löwe et al., 2022; Gopalakrishnan et al., 2024), scene mixtures (Greff et al., 2016; 2017; 2019; Burgess et al., 2019; Engelcke et al., 2019; Locatello et al., 2020; Biza et al., 2023; Didolkar et al., 2023), or layers (Monnier et al., 2021). Notably, Seitzer et al. (2023) leveraged DINO features to extract robust image representations, while Daniel & Tamar (2022; 2023) introduced approaches to disentangle object latents into 2D position and appearance attributes. Despite their efficacy in scene decomposition, they do not model the objects' 3D nature.

**Unsupervised 3D object discovery.** To capture the 3D nature of scenes and objects, recent works have explored learning 3D-aware representations from point clouds (Wang et al., 2022), videos (Henderson & Lampert, 2020), and multi-view images of either single scenes (Liang et al., 2022) or large datasets for generalization (Eslami et al., 2018; Chen et al., 2020; Sajjadi et al., 2022). More recent research focuses on inferring object-centric factorized scene representations from single images (Stelzner et al., 2021; Yu et al., 2022; Smith et al., 2023). Among these, Yu et al. (2022) proposed a method for the unsupervised discovery of object radiance fields (uORF) from single images. Follow-up works (Smith et al., 2023) improved rendering efficiency by replacing NeRF with light fields (Sitzmann et al., 2021) or enhanced segmentation accuracy through bi-level query optimization (Jia et al., 2023). However, these representations often lack translation invariance, which limits their robustness and generalization capabilities. In this work, we address this limitation by incorporating translation invariance, resulting in significant improvements in both generalization and sample efficiency. For a more comprehensive review of related methods, we refer readers to Villa-Vásquez & Pedersoli (2024).

**Object-centric 3D reconstruction.** Decomposing visual scenes on an object-by-object basis and estimating their semantic/geometric attributes has been explored in several recent works (Wu et al., 2017; Yao et al., 2018; Kundu et al., 2018; Ost et al., 2021). Some approaches, such as AutoRF (Müller et al., 2022), successfully reconstruct specific objects (*e.g.*, cars) from annotated images. Others decompose visual scenes into the background and individual objects represented by neural fields (Yang et al., 2021; Wu et al., 2022). Our work differs because of its emphasis on unsupervised learning. Another line of recent work focuses on lifting 2D segmentation to reconstructed 3D scenes (Fan et al., 2022; Cen et al., 2023a;b). In contrast, our work aims at single-image inference, whereas these studies concentrate on multi-view reconstruction.

**Generative neural fields.** Neural fields have revolutionized 3D scene modeling. Early works have shown promising geometric representations (Sitzmann et al., 2019; Park et al., 2019). The seminal work on neural

radiance fields (Mildenhall et al., 2020) has opened up a burst of research on neural fields. We refer the reader to recent survey papers (Tewari et al., 2020; Xie et al., 2022) for a comprehensive overview. In particular, compositional generative neural fields such as GIRAFFE (Niemeyer & Geiger, 2021) and others (Nguyen-Phuoc et al., 2020; Wang et al., 2023b) also allow learning object representations from image collections. Yet, they target unconditional generation and cannot tackle inference.

## 3 Approach

Given a single input image, our goal is to infer object-centric radiance fields (i.e., each discovered object is represented in its local object coordinate rather than the world or the viewer coordinates) and the objects' 3D locations. The object-centric design not only boosts generalizability due to representation invariance, but also allows learning object priors from scenes with different spatial layouts and compositional configurations. The following provides an overview of our approach and then introduces the technical details.

### 3.1 Model Overview

As shown in Figure 2, uOCF consists of an encoder, a latent inference module, and a decoder.

**Encoder.** From an input image $\mathbf{I}$, the encoder extracts a feature map $\mathbf{f} \in \mathbb{R}^{N \cdot C}$, where $N = H \cdot W$ is the spatial size of the feature map and $C$ represents the number of channels. We set it as a frozen DINOv2-ViT (Oquab et al., 2023) followed by two convolutional layers.

**Latent inference module.** The latent inference module infers the latent representation and position of the objects in the underlying 3D scene from the feature map. We assume that the scene is composed of a background environment and no more than $K$ foreground objects. Therefore, the output includes a background latent $\mathbf{z}^{\mathrm{b}} \in \mathbb{R}^{1 \times D}$ and a set of foreground object latent $\mathbf{z}^{\mathrm{f}} = [\mathbf{z}_1^{\mathrm{f}T} \ \mathbf{z}_2^{\mathrm{f}T} \ \cdots \ \mathbf{z}_K^{\mathrm{f}T}]^T \in \mathbb{R}^{K \times D}$ with their corresponding positions $\{\mathbf{p}_i^{\mathrm{wd}}\}_{i=1}^K$, where $\mathbf{p}_i^{\mathrm{wd}} \in \mathbb{R}^3$ denotes a position in the world coordinate. Note that some object latent may be empty when the scene has $< K$ objects.

**Decoder.** Our decoder employs the conditional NeRF formulation $g(\mathbf{x}|\mathbf{z})$, which takes the 3D location $\mathbf{x}$ and the latent $\mathbf{z}$ as input and generates the radiance color and density for rendering. We use two MLPs, $g^{\mathrm{b}}$ and $g^{\mathrm{f}}$, for the background environment and the foreground objects, respectively.

### 3.2 Object-Centric 3D Scene Modeling

**Object-centric latent inference.** Our Latent Inference Module (LIM) aims at binding a set of learnable object queries ( $\mathbf{q}^{\mathrm{f}} = [\mathbf{q}_1^{\mathrm{f}T} \ \mathbf{q}_2^{\mathrm{f}T} \ \cdots \ \mathbf{q}_K^{\mathrm{f}T}]^T \in \mathbb{R}^{K \times D}$ ) to the visual features of each foreground object, and another query to the background features ($\mathbf{q}^{\mathrm{b}} \in \mathbb{R}^{1 \times D}$). The binding is modeled via the cross-attention mechanism with learnable linear functions $\mathcal{K}^{\mathrm{b}}, \mathcal{K}^{\mathrm{f}}, \mathcal{Q}^{\mathrm{b}}, \mathcal{Q}^{\mathrm{f}}, \mathcal{V}^{\mathrm{b}}, \mathcal{V}^{\mathrm{f}}$:

$$\mathbf{A}_{i,j} = \frac{\exp(\mathbf{M}_{i,j})}{\sum_k \exp(\mathbf{M}_{i,k})}, \quad \text{where} \quad \mathbf{M} = \frac{1}{\sqrt{D}} \begin{bmatrix} \mathcal{Q}^{\mathrm{b}}(\mathbf{q}^{\mathrm{b}}) \cdot \mathcal{K}^{\mathrm{b}}(\mathbf{f})^T \\ \mathcal{Q}^{\mathrm{f}}(\mathbf{q}^{\mathrm{f}}) \cdot \mathcal{K}^{\mathrm{f}}(\mathbf{f})^T \end{bmatrix}^T \in \mathbb{R}^{N \times (K+1)}. \tag{1}$$

We then calculate the update signals for queries via an attention-weighted mean of the input:

$$\mathbf{u}^{\mathrm{b}} = (\mathbf{W}_{(:,1)})^T \cdot \mathcal{V}^{\mathrm{b}}(\mathbf{f}) \in \mathbb{R}^{1 \times D}; \quad \mathbf{u}^{\mathrm{f}} = (\mathbf{W}_{(:,2:)})^T \cdot \mathcal{V}^{\mathrm{f}}(\mathbf{f}) \in \mathbb{R}^{K \times D}, \tag{2}$$

where $\mathbf{W}_{i,j} = \frac{\mathbf{A}_{i,j}}{\sum_l \mathbf{A}_{l,j}}$ is the normalized attention map. Queries are then updated by:

$$\mathbf{q}^{\mathrm{b}} \leftarrow \mathbf{q}^{\mathrm{b}} + \mathbf{u}^{\mathrm{b}}, \ \mathbf{q}^{\mathrm{f}} \leftarrow \mathbf{q}^{\mathrm{f}} + \mathbf{u}^{\mathrm{f}}; \quad \mathbf{q}^{\mathrm{b}} \leftarrow \mathbf{q}^{\mathrm{b}} + t^{\mathrm{b}}(\mathbf{q}^{\mathrm{b}}), \ \mathbf{q}^{\mathrm{f}} \leftarrow \mathbf{q}^{\mathrm{f}} + t^{\mathrm{f}}(\mathbf{q}^{\mathrm{f}}), \tag{3}$$

where $t^{\mathrm{b}}$ and $t^{\mathrm{f}}$ are MLPs. We repeat this procedure for $T$ iterations, followed by concatenating the updated object queries with the corresponding attention-weighted mean of the input feature map $\mathbf{f}$ (global residual), finally delivering the background latent $\mathbf{z}^{\mathrm{b}}$ and foreground latent $\{\mathbf{z}_i^{\mathrm{f}}\}_{i=1}^K$.

Our LIM is related to the Slot Attention (Locatello et al., 2020) while differs in several critical aspects. We discuss their relationship in Appendix C.1.

**Object location inference.** To infer objects' position along with their latent representation, we assign a normalized image position $\mathbf{p}_i^{\mathrm{img}} \in [-1, 1]^2$ initialized as zero to each foreground object query, then

Stage 1: **Learn 3D object prior**
from synthetic scenes with simple composition

Stage 2: **Learn to discover objects** from scenes
with diverse object category and spatial layout

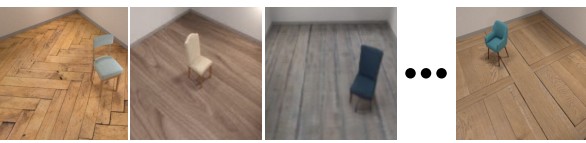 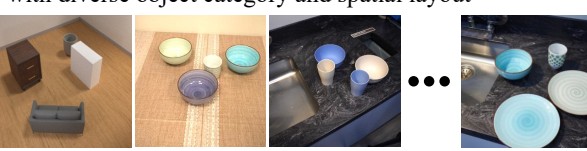

Figure 3: Our object-centric design allows learning 3D object priors that generalize across different scene configurations. We first train our model to learn 3D object priors on simple synthetic scenes (*e.g.*, single synthetic object), and then we leverage the 3D object priors to learn to discover objects in more complex scenes with different object categories and spatial layouts. Note that no object annotation is needed in either stage.

iteratively update them by momentum $m$ with the attention-weighted mean over the normalized 2D grid $\mathbf{E}^{\text{abs}} \in [-1,1]^{N \times 2}$:

$$\mathbf{p}_i^{\text{img}} \leftarrow (\mathbf{W}_{(:,i+1)})^T \cdot \mathbf{E}^{\text{abs}} \cdot (1-m) + \mathbf{p}_i^{\text{img}} \cdot m. \tag{4}$$

To incorporate the inferred positions, we adopt the relative positional encoding (Biza et al., 2023) $\mathbf{E}_i^{\text{pos}} := \text{concat}([\mathbf{E}^{\text{abs}} - \mathbf{p}_i^{\text{img}}, \mathbf{p}_i^{\text{img}} - \mathbf{E}^{\text{abs}}]) \in \mathbb{R}^{N \times 4}, i \in \{1, 2, \cdots, K\}$, and $\mathbf{E}_0^{\text{pos}} := \text{concat}([\mathbf{E}^{\text{abs}}, -\mathbf{E}^{\text{abs}}])$, where concat is the concatenation along the last dimension. Then, we re-write $M$ in Eq. (1) as:

$$M = \frac{1}{\sqrt{D}} \begin{bmatrix} \mathcal{Q}^{\text{b}}(\mathbf{q}^{\text{b}}) \cdot \mathcal{K}^{\text{b}}(\mathbf{f} + h_1(\mathbf{E}_0^{\text{pos}}))^T \\ \mathcal{Q}^{\text{f}}(\mathbf{q}_1^{\text{f}}) \cdot \mathcal{K}^{\text{f}}(\mathbf{f} + h_1(\mathbf{E}_1^{\text{pos}}))^T \\ \cdots \\ \mathcal{Q}^{\text{f}}(\mathbf{q}_K^{\text{f}}) \cdot \mathcal{K}^{\text{f}}(\mathbf{f} + h_1(\mathbf{E}_K^{\text{pos}}))^T \end{bmatrix}^T, \tag{5}$$

where $h_1 : \mathbb{R}^4 \to \mathbb{R}^D$ is a linear function.

Overall, LIM achieves a gradual binding between the queries and the objects in the scene through an iterative update of the queries and their locations. To address potential issues of duplicate object identification, we invalidate one of two similar object queries with high similarity and positional proximity by the start of the last iteration. Finally, a small bias term is added to the position to handle potential occlusion, *i.e.*, $\mathbf{p}_i^{\text{img}} \leftarrow \mathbf{p}_i^{\text{img}} + \tanh(h_2((W_{(:,i+1)})^T)) \cdot \alpha$, where scaling hyperparameter $\alpha = 0.2$ and $h_2 : \mathbb{R}^N \to \mathbb{R}^2$ is a linear function.

The 2D positions $\mathbf{p}_i^{\text{img}}$ are then unprojected into the 3D world coordinate to obtain $\mathbf{p}_i^{\text{wd}}$. To do this, we extend the rays by depth $d \cdot s_i$, where $d$ is the depth estimated by a monocular depth estimator (Ranftl et al., 2022) and $\{s_i\}_{i=1}^K$ are scaling terms predicted by a linear layer using the camera parameters and object latent as input.

**Compositional neural rendering.** The object positions allow us to put objects in their local coordinates rather than the viewer or world coordinates, thereby obtaining object-centric neural fields. Technically, for each 3D point $\mathbf{x}$ in the world coordinate, we transform it to the $i^{\text{th}}$ object's local coordinate by $\mathbf{x}_i = R \cdot (\mathbf{x} - \mathbf{p}_i^{\text{wd}})$, where $R$ denotes the input camera rotation matrix. We then retrieve the color and density of $\mathbf{x}$ in the foreground radiance fields as $(\mathbf{c}_i, \sigma_i) = g^{\text{f}}(\mathbf{x}_i | \mathbf{z}_i^{\text{f}})$ and in the background radiance field as $(\mathbf{c}_0, \sigma_0) = g^{\text{b}}(\mathbf{x} | \mathbf{z}^{\text{b}})$. These values are aggregated into the scene's composite density and color $(\bar{\mathbf{c}}, \bar{\sigma})$ using density-weighted means:

$$\bar{\sigma} = \sum_{i \geq 0} \omega_i \sigma_i, \quad \bar{\mathbf{c}} = \sum_{i \geq 0} \omega_i \mathbf{c}_i, \quad \text{where} \quad \omega_i = \frac{\sigma_i}{\sum_{j \geq 0} \sigma_j}. \tag{6}$$

Finally, we compute the pixel color by volume rendering. Our pipeline is trivially differentiable, allowing backpropagation through all parameters simultaneously.

**Discussion on extrinsics disentanglement.** An object's canonical orientation is ambiguous without assuming its category (Wang et al., 2019). Thus, we choose not to disentangle objects' orientation since we target category-agnostic object discovery. Further, we observe that uOCF has learned meaningful representations that can smoothly interpolate an object's scale and orientation. Please refer to Appendix B for visualization and analysis.

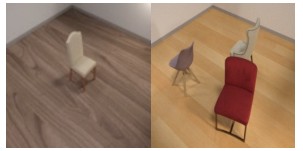 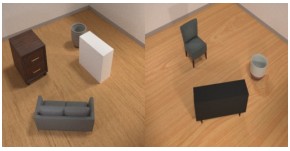 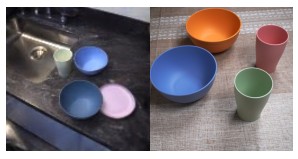 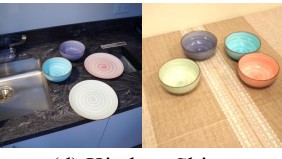

(a) Room-Texture  (b) Room-Furniture  (c) Kitchen-Matte  (d) Kitchen-Shiny

Figure 4: Samples from our collected datasets, where Room-Texture and Room-Furniture consist of synthetic images, and Kitchen-Matte and Kitchen-Shiny consist of real photos.

### 3.3 Object Prior Learning

Unsupervised discovery of 3D objects in complex scenes is inherently difficult due to multiple challenging ambiguities. A major ambiguity is what defines an object. While existing methods define objects via visual appearance similarity (Yu et al., 2022) or priors from 2D segments (Chen et al., 2024), they suffer from under-segmentation due to visual cluttering (Yu et al., 2022) or over-segmentation inherited from the 2D supervision (Chen et al., 2024).

We explore addressing this challenge by learning 3D object priors from synthetic data. Existing methods have difficulties learning generalizable 3D object priors, as their object representation is sensitive to spatial configurations: a minor shift in camera pose or object location, rather than the object itself, can lead to drastic changes in the object representation. Thus, such learned object priors do not generalize when there are unseen spatial configurations.

Our 3D object-centric representation mitigates this issue by translation invariance. In particular, we introduce 3D object prior learning. We show an illustration in Figure 3. The main idea is to pre-train uOCF on synthetic scenes that are constructed with a single object to ease the learning, similar to curriculum learning. After the pre-training stage, we proceed to training uOCF on the more complex scenes that may have different object categories and spatial layouts. Note that either training stage does not require any object annotation. The pre-training synthetic single-object dataset can be easily scaled up.

### 3.4 Model Training

**Object-centric sampling.** To improve the reconstruction quality, we leverage an object's local coordinates to concentrate the sampled points in proximity to the object. Specifically, we start dropping distant samples from the predicted object positions after a few training epochs when the model has learned to distinguish the foreground objects and predict their positions. This approach enables us to quadruple the number of samples with the same amount of computation, leading to significantly improved robustness and visual quality.

In both training stages, we train our model across scenes, each with calibrated sparse multi-view images. For each training step, the model receives an image as input, infers the objects' latent representations and positions, renders multiple views from the input and reference poses, and compares them to the ground truth images to calculate the loss. Model supervision consists of the MSE reconstruction loss $\ell_{\text{recon}}$ and the perceptual loss $\ell_{\text{perc}}$ (Johnson et al., 2016) between the reconstructed and ground truth images. In addition, we incorporate the depth ranking loss (Wang et al., 2023a) with pre-trained monocular depth estimators and background occlusion regularization (Yang et al., 2023) to minimize common floating artifacts in few-shot NeRFs.

The overall loss function is thus formulated as follows:

$$\mathcal{L} = \ell_{\text{recon}} + \lambda_{\text{perc}}\ell_{\text{perc}} + \lambda_{\text{depth}}\ell_{\text{depth}} + \lambda_{\text{occ}}\ell_{\text{occ}}. \tag{7}$$

We leave further architectural details and illustrations in Appendix C.1.

## 4 Experiments

We evaluate our method on three tasks: unsupervised object segmentation in 3D, novel view synthesis, and scene manipulation in 3D. Below, we briefly describe the data collection process and experimental configurations, with additional details provided in Appendices C.2 and C.3. Sample code and data are included in the supplementary material, and we plan to release the full code and datasets for public use.

Table 1: Object segmentation and view synthesis on Room-Texture and Room-Furniture.

| Method | Room-Texture | | | | | | Room-Furniture | | | | | |
| | Object segmentation | | | Novel view synthesis | | | Object segmentation | | | Novel view synthesis | | |
| | ARI↑ | FG-ARI↑ | NV-ARI↑ | PSNR↑ | SSIM↑ | LPIPS↓ | ARI↑ | FG-ARI↑ | NV-ARI↑ | PSNR↑ | SSIM↑ | LPIPS↓ |
|---|---|---|---|---|---|---|---|---|---|---|---|---|
| uORF (Yu et al., 2022) | 0.670 | 0.093 | 0.578 | 24.23 | 0.711 | 0.254 | 0.686 | 0.497 | 0.556 | 27.49 | 0.780 | 0.258 |
| BO-QSA (Jia et al., 2023) | 0.697 | 0.354 | 0.604 | 25.26 | 0.739 | 0.215 | 0.682 | 0.479 | 0.579 | 27.29 | 0.774 | 0.261 |
| COLF (Smith et al., 2023) | 0.235 | 0.532 | 0.011 | 22.98 | 0.670 | 0.504 | 0.514 | 0.458 | 0.439 | 28.73 | 0.781 | 0.386 |
| uOCF (ours) | **0.785** | **0.563** | **0.704** | **28.85** | **0.798** | **0.136** | **0.861** | **0.739** | **0.808** | **29.77** | **0.830** | **0.127** |

Figure 5: Scene segmentation qualitative results. Novel view images are for reference only.

**Data.** We collect two synthetic datasets and two real-world datasets to evaluate our method. Examples of these datasets are shown in Figure 4.

Room-Texture. Room-Texture features 324 object models from the "armchair" category of the ABO(Collins et al., 2022) dataset. Each scene includes 2–4 objects arranged on backgrounds randomly selected from a collection of floor textures. The dataset comprises 5,000 scenes for training and 100 for evaluation. Each scene is rendered from four viewpoints centered on the scene.

Room-Furniture. In Room-Furniture, objects are selected from 1,425 ABO (Collins et al., 2022) models spanning seven categories, including "bed", "cabinet", "chair", "dresser", "ottoman", "sofa", and "plant pot". Other configurations match that of Room-Texture.

Kitchen-Matte. This dataset includes scenes featuring single-color matte dinnerware set against two types of backgrounds: a plain tabletop or a complex kitchen environment. The dataset contains 735 scenes for training and 102 for evaluation. Each scene includes 3–4 objects positioned randomly and is captured from three viewpoints (for tabletop scenes) or two viewpoints (for kitchen backdrops).

Kitchen-Shiny. This dataset contains scenes with textured, shiny dinnerware. Similar toKitchen-Matte, the first half of the dataset features a plain tabletop, while the latter half includes a kitchen background. The dataset consists of 324 scenes for training and 56 for evaluation.

Table 2: Novel view synthesis on Kitchen-Shiny and Kitchen-Matte.

| Method | Kitchen-Shiny | | | Kitchen-Matte | | |
|---|---|---|---|---|---|---|
| | PSNR↑ | SSIM↑ | LPIPS↓ | PSNR↑ | SSIM↑ | LPIPS↓ |
| uORF (Yu et al., 2022) | 19.23 | 0.602 | 0.336 | 26.07 | 0.808 | 0.092 |
| BO-QSA (Jia et al., 2023) | 19.78 | 0.639 | 0.318 | 27.36 | 0.832 | 0.067 |
| COLF (Smith et al., 2023) | 18.30 | 0.561 | 0.397 | 20.68 | 0.643 | 0.236 |
| uOCF (ours) | **28.58** | **0.862** | **0.049** | **29.40** | **0.867** | **0.043** |

Table 3: Novel view synthesis on Kitchen-Shiny with a larger number of object queries $K$.

| | PSNR↑ | SSIM↑ | LPIPS↓ |
|---|---|---|---|
| $K = 4$ | 28.58 | 0.862 | 0.049 |
| $K = 5$ | 28.28 | 0.846 | 0.059 |
| $K = 6$ | 28.04 | 0.848 | 0.058 |
| $K = 10$ | 28.20 | 0.840 | 0.065 |

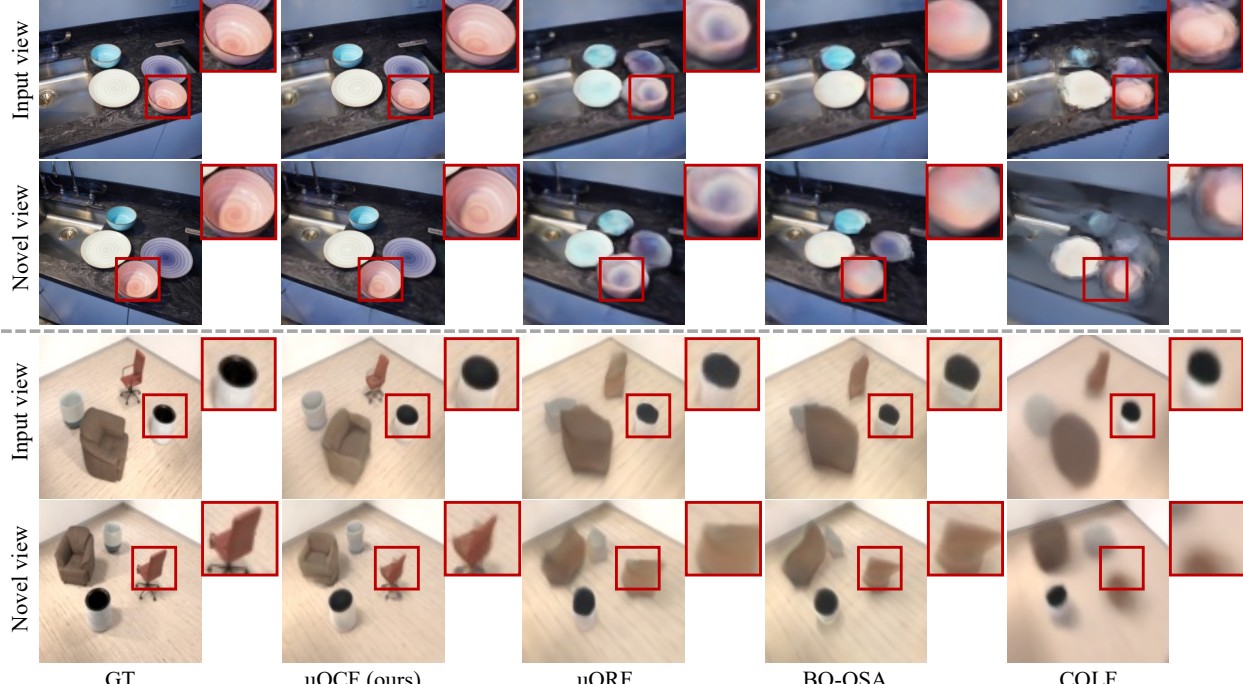

Figure 6: Novel view synthesis qualitative results on Kitchen-Shiny (top) and Room-Furniture (bottom).

**Implementation details.** To learn object priors, we generate a synthetic dataset of over 8,000 scenes. Each scene contains one object, sampled from a high-quality subset of Objaverse-LVIS (Deitke et al., 2023), placed against a room background. These objects span over 100 categories. The synthetic dataset is easy to generate and scalable, making it ideal for learning object priors for all our experiments.

In the second stage, the number of foreground object queries is set to $K = 4$. We initialize the model with the pre-trained weights from the object prior learning stage and train it on multi-object scenes. Once trained, our model can perform direct inference on images with spatial configurations that differ from those seen during training. Additionally, our model can adapt to unseen environments through efficient test-time optimization (see Sec. 4.2 for details).

**Baselines.** We compare our method against uORF (Yu et al., 2022), BO-QSA (Jia et al., 2023), and COLF (Smith et al., 2023). To ensure a fair comparison, we increase the latent dimensions and training iterations for all methods. For baseline models, we retain their original implementation without incorporating our proposed object-centric learning stage to maintain consistency. Details on baselines enhanced with our object-centric learning stage can be found in Appendix D.

**Metrics.** We report the PSNR, SSIM, and LPIPS metrics for novel view synthesis. For scene segmentation, we use three variants of the Adjusted Rand Index (ARI): the conventional ARI (calculated on all input image pixels), the Foreground ARI (FG-ARI, calculated on foreground input image pixels), and the Novel View ARI (NV-ARI, calculated on novel view pixels). All scores are computed on images of resolution $128 \times 128$.

Table 4: Scene manipulation results on the Room-Texture dataset.

| Method | Object Translation | | | Object Removal | | |
|---|---|---|---|---|---|---|
| | PSNR↑ | SSIM↑ | LPIPS↓ | PSNR↑ | SSIM↑ | LPIPS↓ |
| uORF (Yu et al., 2022) | 23.65 | 0.654 | 0.284 | 23.81 | 0.664 | 0.282 |
| BO-QSA (Jia et al., 2023) | 25.21 | 0.700 | 0.226 | 24.58 | 0.698 | 0.247 |
| uOCF (ours) | **27.66** | **0.774** | **0.156** | **28.99** | **0.802** | **0.136** |

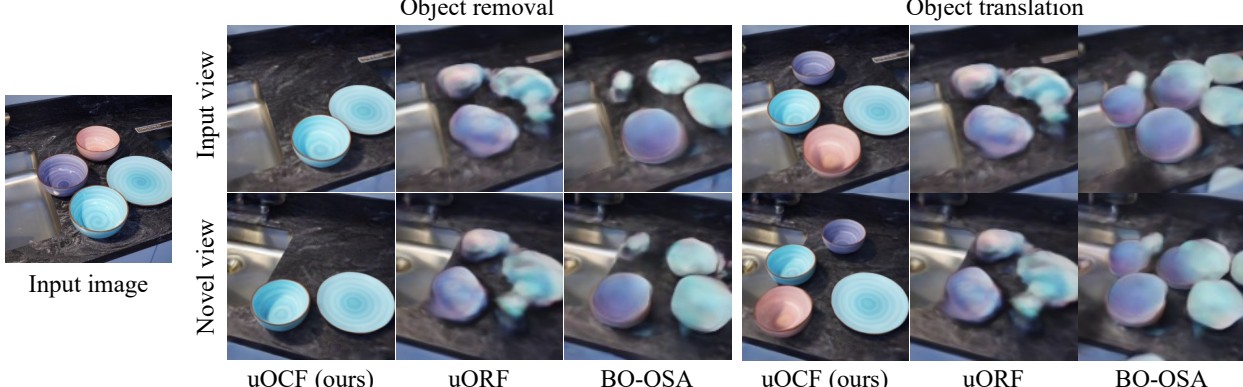

Figure 7: Qualitative results of single-image 3D scene manipulation on the Kitchen-Shiny dataset.

## 4.1 Baseline Comparison on Multiple Tasks

**Unsupervised object segmentation in 3D.** We evaluate the object discovery quality by object segmentation in 3D. We render a density map $\mathbf{d}^i$ for each latent $i$ and assign each pixel $p$ a segmentation label $s_p = \arg\max_{i=0}^{K} \mathbf{d}_p^i$ in the input view and novel views. We show our results in Table 1 and examples in Figure 5. From Table 1, we see that our uOCF outperforms all existing methods in all metrics. From Figure 5, we observe that no prior method can produce reasonable segmentation results in real-world Kitchen-Shiny scenes. Specifically, uORF binds all objects to the background, resulting in empty object segmentation; BO-QSA fails to distinguish different object instances; COLF produces meaningless results on novel views. A fundamental issue in these methods is that they lack appropriate object priors to handle the ambiguity in disentangling multiple objects. In contrast, uOCF can discover objects in real-world scenes. Moreover, uOCF can handle scenes where objects occlude each other. We provide more visualization results in Appendix D.

**Novel view synthesis.** We evaluate the scene and object reconstruction quality by novel view synthesis. For each test scene, we use a single image as input and other views as references. We show our results in Table 2 and examples in Figure 6. We also show additional results in Appendix D. Our method significantly surpasses the baselines in all metrics. Importantly, while previous methods often fail to distinguish foreground objects and thus produce blurry reconstruction of objects, our approach consistently produces high-fidelity scene and object reconstruction and novel view synthesis results.

**Scene manipulation in 3D.** We further evaluate object discovery by single-image 3D scene manipulation. Since uOCF explicitly infers 3D locations of discovered objects, it readily supports: 1) object translation by modifying an object's position, and 2) object removal by excluding objects during compositional rendering.

For quantitative evaluation, we create a test set by randomly selecting an object in each of the Room-Texture scenes, and shift its position (object translation) or remove it (object removal). During inference, we determine the object to manipulate by selecting the object with the highest IoU score with the ground truth mask. As shown in Table 4, uOCF outperforms baselines across all metrics in both object translation and object removal due to its better performance in object discovery. We further show qualitative examples from the Kitchen-Shiny dataset in Figure 7. We observe that uORF merges all objects into the background, and thus the manipulation results are identical to the original reconstruction; BO-QSA fails to distinguish foreground objects, resulting in blurry manipulation results (we show more visualization in Appendix D).

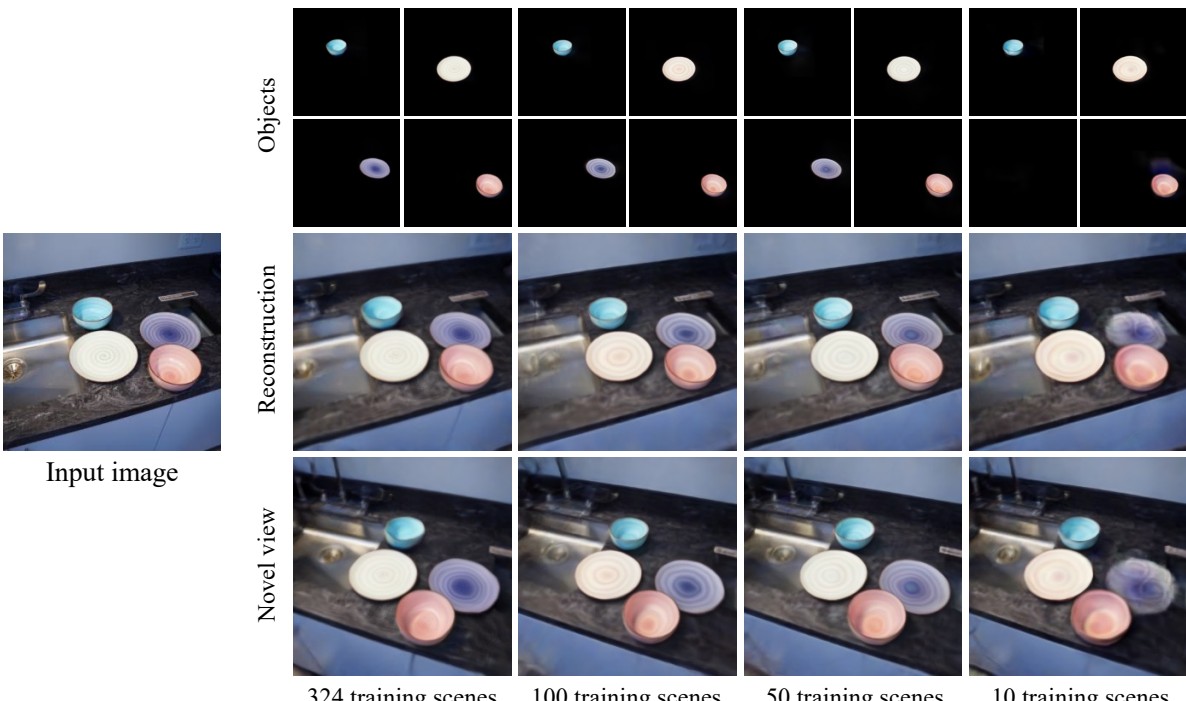

Figure 8: Qualitative results on sample efficiency. With fewer training scenes, our uOCF can still produce reasonable object discovery thanks to the object-centric modeling and learned object priors.

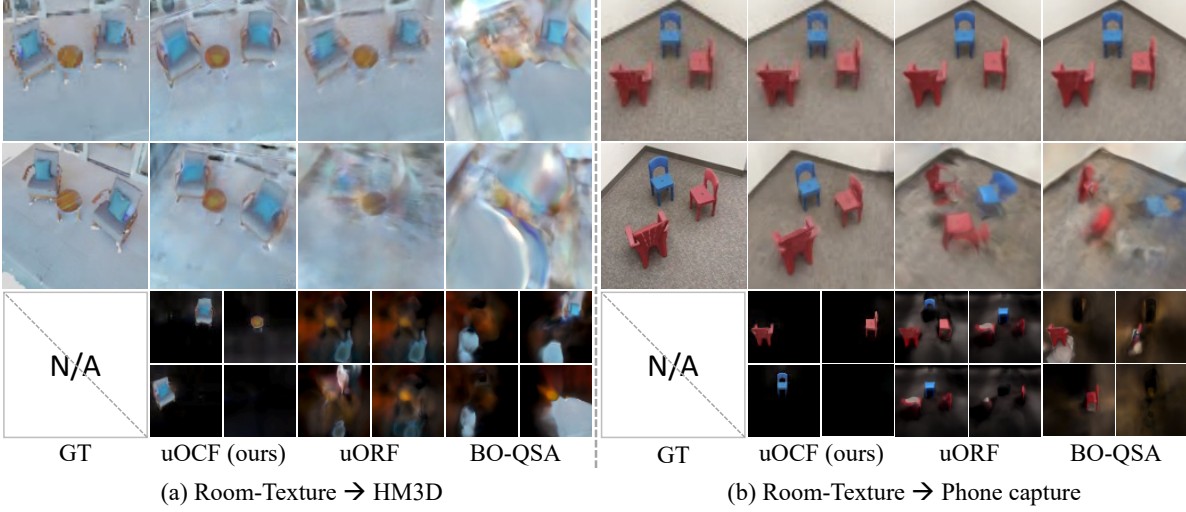

Figure 9: Zero-shot generalization results. We load the model trained on one dataset and test it on an image from another dataset after a fast test-time optimization on the *input view only*. First/second/third row: scene reconstruction/novel view/objects.

In contrast, our uOCF delivers much higher-quality manipulation results. We show additional visualization results in the supplementary video.

## 4.2 Generalization Analysis

In the experiments above, all test scenes have unseen novel spatial configurations, where uOCF shows strong generalization. We further evaluate the sample efficiency on spatial generalization, and we showcase the generalization to unseen objects.

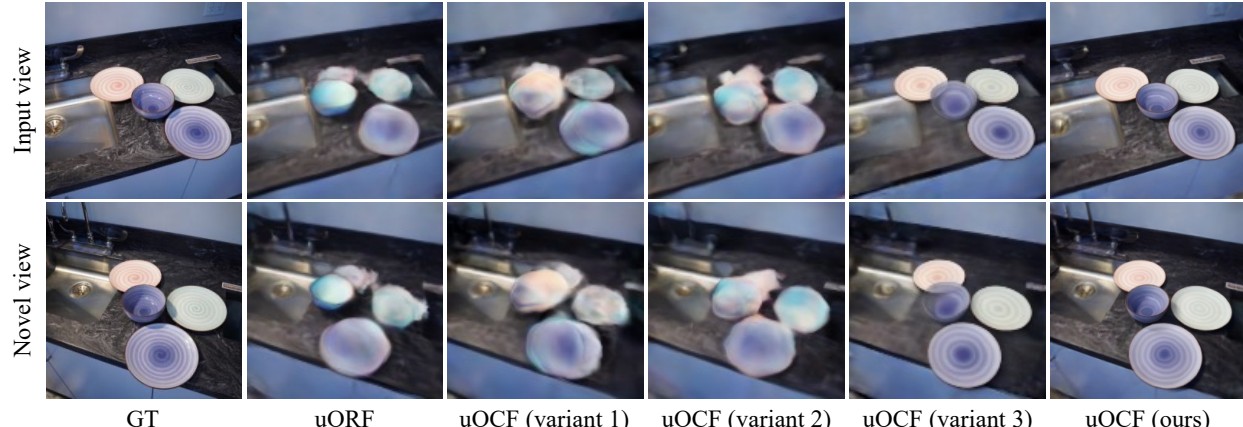

Figure 10: Ablation on translation invariance and object prior learning. Three variants: (1) without both translation invariance and object prior learning, (2) without translation invariance, (3) without object prior learning.

**Sample efficiency.** We train uOCF with a small subset of (e.g., only 10) the training scenes, and test it on the test set. As shown by the qualitative example in Figure 8, even when we only have a few training scenes, uOCF still demonstrates a good generalization ability to discover objects. This is mainly due to the translation invariance and learned object priors, which reduce the dependence on massive training scenes.

**Generalization to unseen objects.** We evaluate the zero-shot generalization ability of uOCF by training it on one dataset and test it on a single image of unseen background and objects. Specifically, we test our model on five real-world examples (one from the HM3D dataset (Ramakrishnan et al., 2021) and four captured with a phone) using a model trained solely on the synthetic Room-Texture dataset. As shown in Figures 9 and , existing methods struggle to adapt to novel objects in unseen settings. In contrast, uOCF demonstrates remarkable generalizability, requiring only a lightweight single-image test-time optimization for 1000 iterations. This process takes approximately 3 minutes, a fraction of the 6 days needed for the full training of the model. These results highlight uOCF 's ability to adapt effectively from synthetic training data to real-world scenarios. Details on this experiment can be found in Appendix C.3.

## 4.3 Ablation Study

**Key technical contributions.** We conduct ablation studies to analyze the impact of our key technical contributions: the translation-invariant design and object prior learning.

As shown in Table 5 and Figure 10, incorporating translation invariance dramatically reduces the LPIPS metric from 0.186 to 0.049. Similarly, leveraging object prior learning significantly decreases LPIPS from 0.125 to 0.049. These results demonstrate that both contributions are not only individually impactful but also complementary. Removing either one severely degrades performance, justifying their importance in achieving high-quality results.

**Other technical improvements.** We also evaluate the impact of other technical improvements through ablation studies. As shown in Table 6, the inclusion of DINO ViT and standard attention enhances overall performance. However, these components contribute relatively modestly; for instance, removing DINO or standard attention slightly increases LPIPS from 0.049 to 0.060 or 0.062, respectively. Excluding depth and occlusion losses also degrades visual quality, leading to a noticeable drop in performance. Additionally, removing the object-centric sampling strategy slightly reduces overall reconstruction quality, further highlighting the significance of these improvements.

**Different $K$ values.** We evaluate the effectiveness of different $K$ values and different scales of data used for 3D object prior learning. We evaluate our method's robustness to different $K$ values, and we show results in Table 3 and Figure 11. From Table 3, we can see that even when we set $K = 10$ which is much higher than the number of possible maximal objects (i.e., 4), our model is robust and gives comparable results. From Figure 11, we observe that even if there are more object queries than the number of objects in the scene, uOCF learns to generate "empty" object queries instead of over-segmenting the objects.

Table 5: Ablation studies on our key technical contributions on Kitchen-Shiny.

| Method | PSNR ↑ | SSIM ↑ | LPIPS ↓ |
|---|---|---|---|
| w/o trans. invar. or object prior learning | 20.68 | 0.645 | 0.303 |
| w/o trans. invar. | 23.70 | 0.724 | 0.186 |
| w/o object prior learning | 26.81 | 0.806 | 0.125 |
| uOCF (ours) | **28.58** | **0.862** | **0.049** |

Table 6: Ablation study on model architecture and loss function on Kitchen-Shiny.

| Method | PSNR↑ | SSIM↑ | LPIPS↓ |
|---|---|---|---|
| w/o DINO | 26.25 | 0.831 | 0.060 |
| w/o standard attention | 27.82 | 0.844 | 0.062 |
| w/o object-centric sampling | 27.31 | 0.852 | 0.072 |
| w/o $\ell_{depth}$ and $\ell_{occ}$ | 26.79 | 0.819 | 0.081 |
| uOCF (ours) | **28.58** | **0.862** | **0.049** |

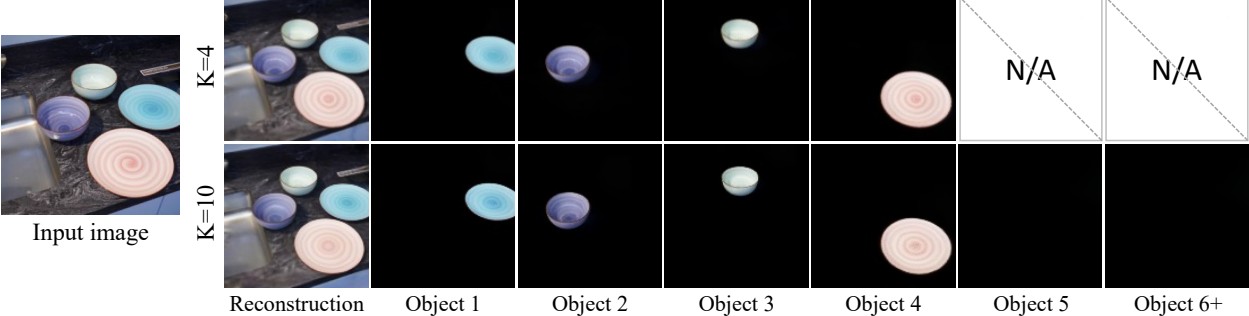

Figure 11: Qualitative results of uOCF on scenes with larger object queries $K$. The order of the object reconstructions is rearranged for better visualization.

## 5  Conclusion

We study the importance of translation invariance for unsupervised 3D object discovery, instantiated as our model for the unsupervised discovery of Object-Centric neural Fields (uOCF). Our results show that our translation-invariant design and the 3D object prior learning can substantially improve the spatial generalization and sample efficiency. Our results demonstrate that unsupervised 3D object discovery can be extended to real scenes while obtaining satisfactory performances.

**Limitations.** Although uOCF shows promising unsupervised 3D object discovery results, it is currently limited to simple real scenes such as the kitchen scenes. Extending to more complex real scenes with complex spatial layouts and a large number of objects from different categories is an important future direction. We leave more discussion on technical limitations in Appendix E.

### Acknowledgments

This work is in part supported by NSF RI #2211258 and #2338203, ONR MURI N00014-22-1-2740, and ONR YIP N00014-24-1-2117.

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

# A  Appendix Overview

This supplementary document is structured as follows: We begin with the proof of concept in Appendix B and provide the implementation details in Appendix C. Then, we discuss the limitations of our approach in Appendix E and present additional qualitative results in Appendix D. Accompanying this document is our *project page with an overview video* attached in the supplementary file.

# B  Proof of Concept

We conduct a toy experiment (Figure 12) to demonstrate that our model has successfully learned object position, rotation, and scale. In this experiment, we begin with two images (input 1 and input 2) of a chair placed at the scene's center, exhibiting different sizes (on the left) or rotation angles (on the right), all captured from the same viewing direction.

We extract the object latents from these images, interpolate them, and then send the interpolated latents to the decoder. As shown between the two input images, we observe a smooth transition in object size and rotation, indicating that the latent representation has effectively captured the scale and rotation of objects.

In the second row, we placed the chairs in different positions. As shown on the right, we obtained a smooth transition again, proving that our model could disentangle object positions from the latent representation.

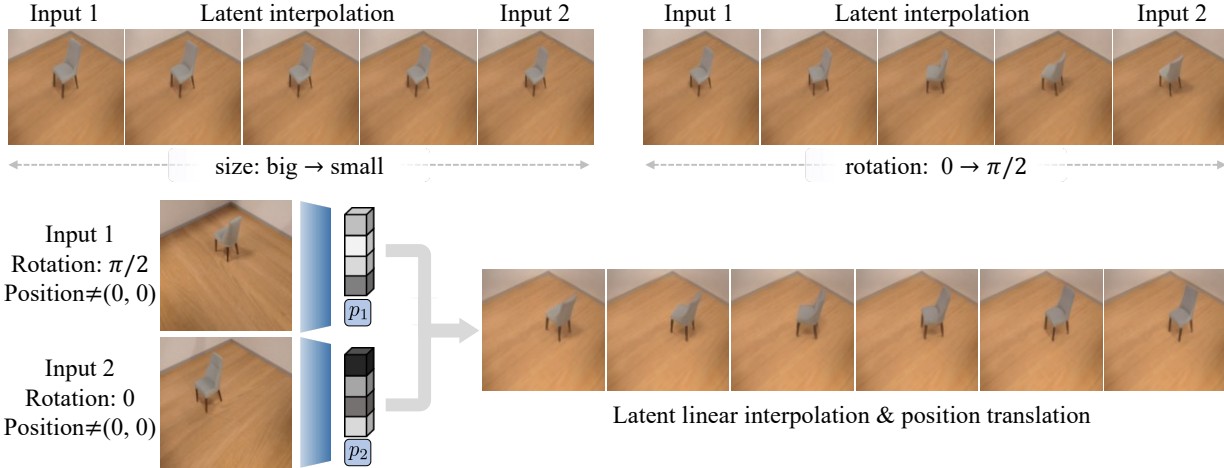

Figure 12: Proof of concept. We demonstrate that uOCF has effectively learned objects' scale and orientation along with the translation-invariant object representation by interpolating the representation of two identical objects with different orientations and scales to obtain transitional results.

# C  Implementation

## C.1  Model Architecture

**Encoder.** Our encoder module consists of a frozen DINO encoder and two convolutional layers. We illustrate its architecture in Figure 13(a).

**Latent inference module.** While motivated by the background-aware slot attention module proposed by (Yu et al., 2022), our latent inference module exhibits three key differences: (1) The object queries are initialized with learnable embeddings instead of being sampled from learnable Gaussians, which enhances training stability; (2) We jointly extract object positions and their latent representations and add object-specific positional encoding to utilize the extracted position information; (3) We remove the Gated Recurrent Unit (GRU) and replace it with the transformer architecture to smooth the gradient flow.

## C.2  Data Collection

This section introduces the details of our datasets.

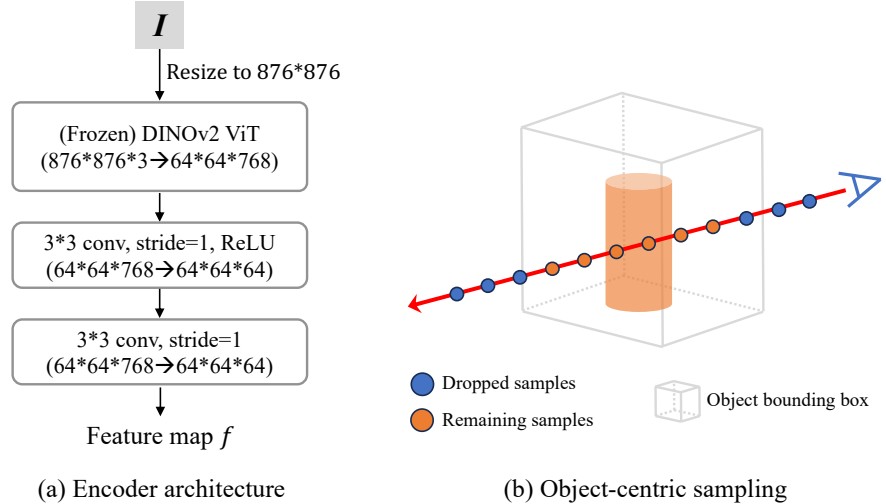

(a) Encoder architecture          (b) Object-centric sampling

Figure 13: (a) Architecture of our encoder module. (b) Object-centric sampling: We drop the samples distant from the predicted object position for efficient sampling..

Room-Texture. In Room-Texture, objects are chosen from 324 ABO objects (Collins et al., 2022) from the "armchair" category. The single-object subset contains four scenes for each object instance, resulting in 1296 scenes in total. The multiple-object subset includes $5,000$ scenes for training and 100 for evaluation, with each scene containing 2-4 objects set against a background randomly chosen from a collection of floor textures. Each scene is rendered from 4 directions toward the center.

Room-Furniture. In Room-Furniture, objects are chosen from $1,425$ ABO (Collins et al., 2022) object models, spanning across seven categories, including "bed", "cabinet", "chair", "dresser", "ottoman", "sofa", and "plant pot". Each scene contains 2-4 objects set against a background randomly chosen from a collection of floor textures. We render 5000 scenes for training and 100 scenes for evaluation.

Kitchen-Matte. In Kitchen-Matte, objects are diffuse and have no texture. The dataset comprises 16 objects and 6 tablecloths in total. We captured 3 images for each tabletop scene and 2 for each kitchen scene. This dataset contains 735 scenes for training and 102 for evaluation, each containing 3-4 objects. We calibrate the cameras using the OpenCV library.

Kitchen-Shiny. In Kitchen-Shiny, objects are specular, and the lighting is more complex. The dataset comprises 12 objects and 6 tablecloths, and the other settings are identical to Kitchen-Matte. This dataset contains 324 scenes for training and 56 for evaluation, each containing 4 objects.

### C.3 Training Configuration

This section discusses the training configuration of uOCF.

We employ Mip-NeRF (Barron et al., 2021) as our NeRF backbone and estimate the depth maps by MiDaS (Ranftl et al., 2022). An Adam optimizer with default hyper-parameters and an exponential decay scheduler is used across all experiments. The initial learning rate is 0.0003 for the first stage and 0.00015 for the second stage. Loss weights are set to $\lambda_{\text{perc}} = 0.006, \lambda_{\text{depth}} = 1.5,$ and $\lambda_{\text{occ}} = 0.1$. The position update momentum $m$ is set to 0.5, and the latent inference module lasts $T = 6$ iterations. Most hyperparameters are inherited from Yu et al. (2022), while the loss weights of our proposed losses are chosen to ensure a similar scale between all loss terms. All experiments are run on a single RTX-A6000 GPU. Training lasts approximately 6 days, including 1.5 days for object prior learning and another 4.5 days for training on multi-object scenes.

**Coarse-To-Fine progressive training.** We employ a coarse-to-fine strategy in our second training stage to facilitate training at higher resolutions. Reference images are downsampled to a lower resolution ($64 \times 64$)

during the coarse training stage and replaced by image patches with the same size as the low-resolution images randomly cropped from the high-resolution ($128 \times 128$) input images during the fine training stage.

**Locality constraint and object-centric sampling.** We employ the locality constraint (a bounding box for foreground objects in the world coordinate) proposed by (Yu et al., 2022) in both training stages but only adopt it before starting object-centric sampling. The number of samples along each ray before and after starting object-centric sampling is set to 64 and 256, respectively. We provide an illustration of our object-centric sampling strategy in Figure 13(b).

**Training configuration on Room-Texture**. During stage 1, we train the model for 100 epochs directly on images of resolution $128 \times 128$. We start with the reconstruction loss only, add the perceptual $10^{\text{th}}$ epoch, and start the object-centric sampling at the $20^{\text{th}}$ epoch. During stage 2, we train the model for 60 epochs on the coarse stage and another 60 on the fine stage. We start with the reconstruction loss only, add the perceptual loss at the $10^{\text{th}}$ epoch, and start the object-centric sampling from the $20^{\text{th}}$ epoch.

**Training configuration on Kitchen-Matte and Kitchen-Shiny**. Both kitchen datasets share the same training configuration with Room-Texture in stage 1. During stage 2, we train the model for 750 epochs, where the fine stage starts at the $250^{\text{th}}$ epoch. We add the perceptual loss at the $50^{\text{th}}$ epoch and start the object-centric sampling from the $150^{\text{th}}$ epoch.

**Training configuration for zero-shot generalization.** For the test-time adaptation, we fine-tune our model on the input view only using our proposed loss function (Eq. 7) for 1000 iterations on resolution $128 \times 128$. We use the Adam optimizer and the learning rate is set to $1 \times 10^{-4}$. This optimization takes about 3 minutes on a single A6000 gpu.

## D   Additional Experiments

**Additional real-world dataset.** We introduce an additional real-world dataset, named "Planters," which features tabletop scenes containing four plant pots or vases arranged on tablecloths. The dataset includes 745 scenes for training and 140 scenes for evaluation, with each scene captured from three different camera poses. As shown in the quantitative results in Table 7 and the qualitative results in Figure 17, our method significantly outperforms existing approaches. It achieves superior scene reconstruction and novel view synthesis, delivering results with noticeably higher visual quality.

| Method | PSNR↑ | SSIM↑ | LPIPS↓ |
|---|---|---|---|
| uORF | 24.49 | 0.748 | 0.163 |
| uORF-BO-QSA | 28.09 | 0.847 | 0.108 |
| COLF | 19.22 | 0.588 | 0.464 |
| uOCF (ours) | **29.00** | **0.864** | **0.062** |

Table 7: Quantitative results on the Planters dataset.

**Baseline performance with object-centric learning.** To ensure fair comparisons, we conducted additional experiments where object prior learning was incorporated into the baseline methods. The results in Table 8 demonstrate that even with the incorporation of object prior learning, uOCF significantly outperforms existing methods due to its translation-invariant object representation, which enhances generalization and data efficiency.

**Additional zero-shot evaluation.** We evaluate our method on two additional phone-captured scenes to further demonstrate the generalizability of uOCF. To ensure fair comparisons, we incorporate object prior learning into the baseline methods. However, as shown in Figure 15, the baseline methods still struggle generalizing to unseen environments. In contrast, our method produces accurate object segmentation and novel view synthesis results, further validating its effectiveness.

**Visualization on discovered objects.** We visualize the discovered objects in Figure 16. Notably, uORF (Yu et al., 2022) puts all objects within the background, whereas BO-QSA (Jia et al., 2023) binds the same object to all queries, resulting in identical foreground reconstruction. In contrast, uOCF accurately differentiates between the foreground objects and the background.

| Method | LPIPS ↓ | SSIM ↑ | PSNR ↑ |
|---|---|---|---|
| uORF | 0.336 | 0.602 | 19.23 |
| uORF + object prior learning | 0.193 | 0.714 | 22.78 |
| BO-QSA | 0.318 | 0.639 | 19.78 |
| BO-QSA + object prior learning | 0.129 | 0.766 | 24.00 |
| COLF | 0.397 | 0.561 | 18.30 |
| COLF + object prior learning | 0.290 | 0.709 | 21.66 |
| uOCF (ours) | 0.049 | 0.862 | 28.58 |

Table 8: Baseline performance with object-centric learning. Our method maintains its superiority even when baselines methods employ our proposed object-prior learning approach due to its translation-invariance representation.

**Visualization on object segmentation in 3D.** We show scene segmentation results on the kitchen datasets in Figure 18. Unlike compared methods that yield cluttered results, uOCF consistently yields high-fidelity segmentation results.

**Additional novel view synthesis results.** We show more qualitative results for novel view synthesis in Figures 18, 19, and 20. Our method produces much better results than compared methods regarding visual quality.

# E Limitations Analysis

**Limitation on reconstruction quality.** Scene-level generalizable NeRFs (Yu et al., 2021; Sajjadi et al., 2022; Yu et al., 2022) commonly face challenges in accurately reconstructing detailed object textures. Our approach also has difficulty capturing extremely high-frequency details. As shown in Figure 14(a), our method fails to replicate the mug's detailed texture. Future research may benefit from stronger object priors learned from larger-scale datasets, such as Large Reconstruction Models (Hong et al., 2023).

**Failure in position prediction.** Our two-stage training pipeline, despite its robustness in many situations, is not immune to errors, particularly in object position prediction. Due to the occlusion between objects, using the attention-weighted mean for determining object positions can sometimes lead to inaccuracies. Although a bias term can rectify this in most instances (Figure 6), discrepancies persist under a few conditions, as depicted in Figure 14(b).

**Training instability.** Like other slot-based object discovery methods, our approach also faces challenges with training instability. For example, the model occasionally collapses within the first few training epochs, even when using identical hyperparameters. To address this, we perform multiple trials for each experiment, terminating early if signs of collapse appear during the initial training stages. We observed that approximately half of the experiments fail at this stage. However, once the model progresses beyond this critical phase, the final results remain consistent across different trials. For baseline methods that struggle on our proposed complex datasets, we conduct at least five trials to ensure that their failure stems from limitations in their design rather than issues with random initialization.

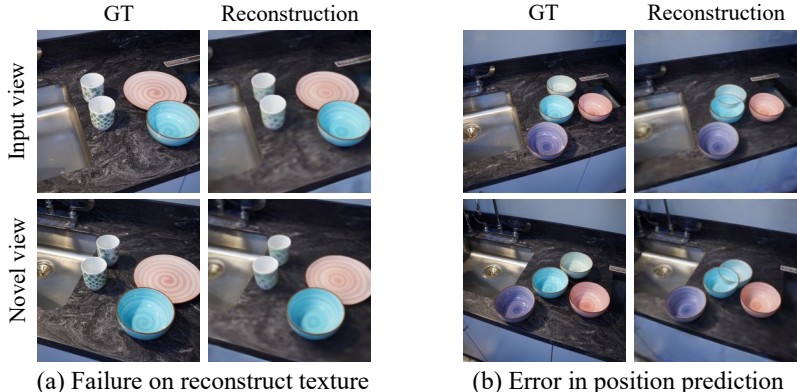

(a) Failure on reconstruct texture    (b) Error in position prediction

Figure 14: Failure case visualizations. Our method may fail to reconstruct intricate object texture or predict biased object position.

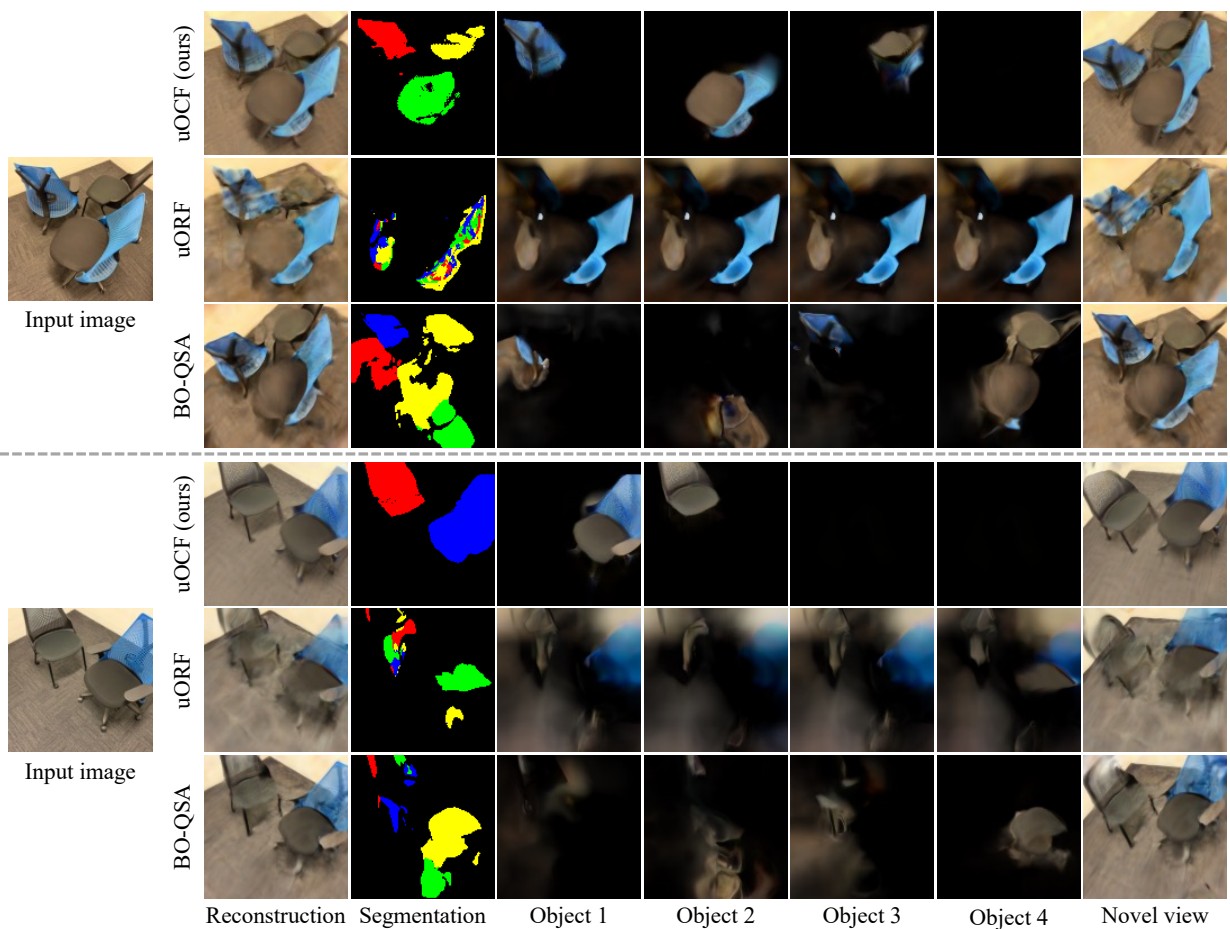

Figure 15: Qualitative zero-shot generalization results.

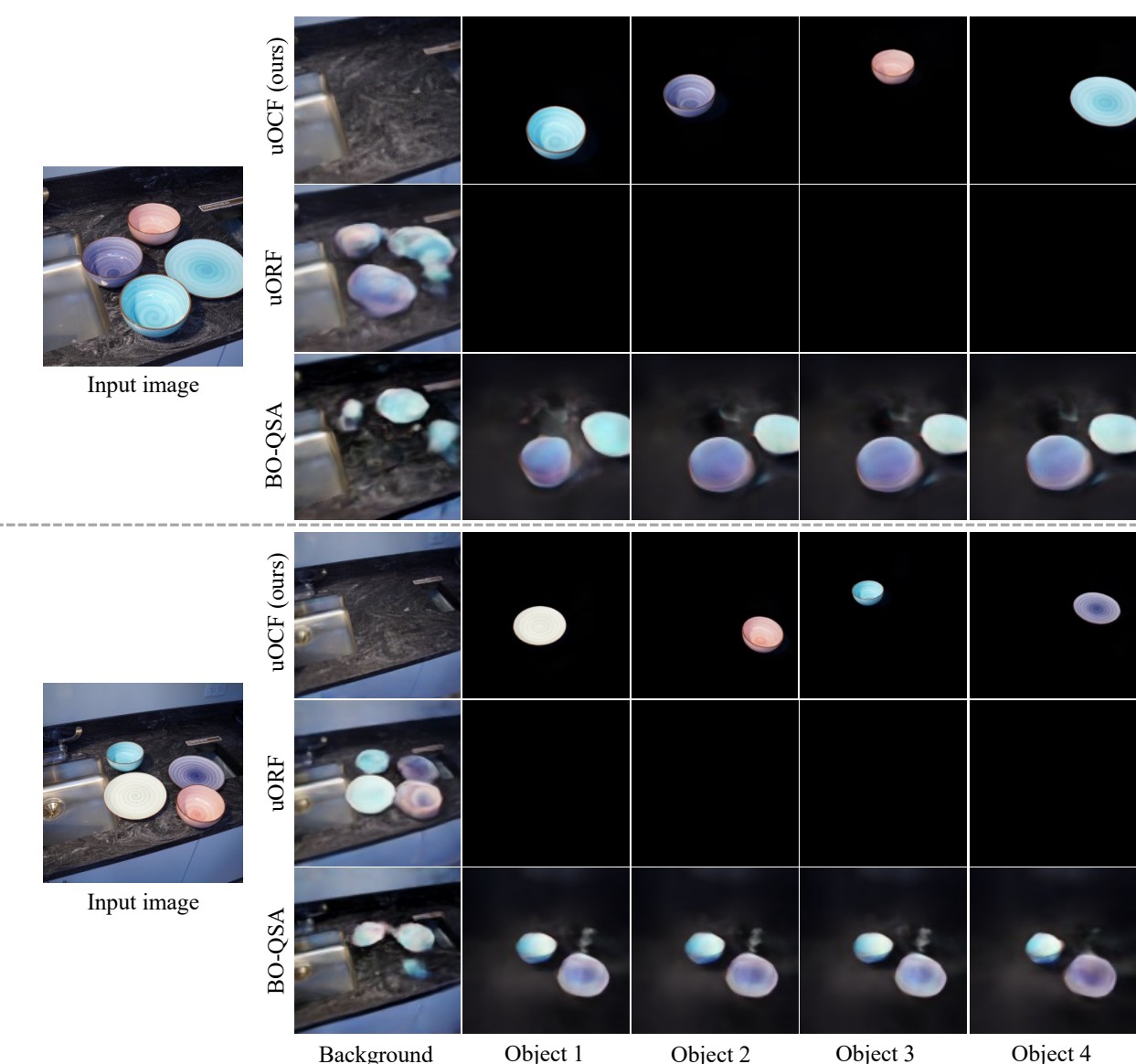

Figure 16: Visualization on discovered objects on Kitchen-Shiny.

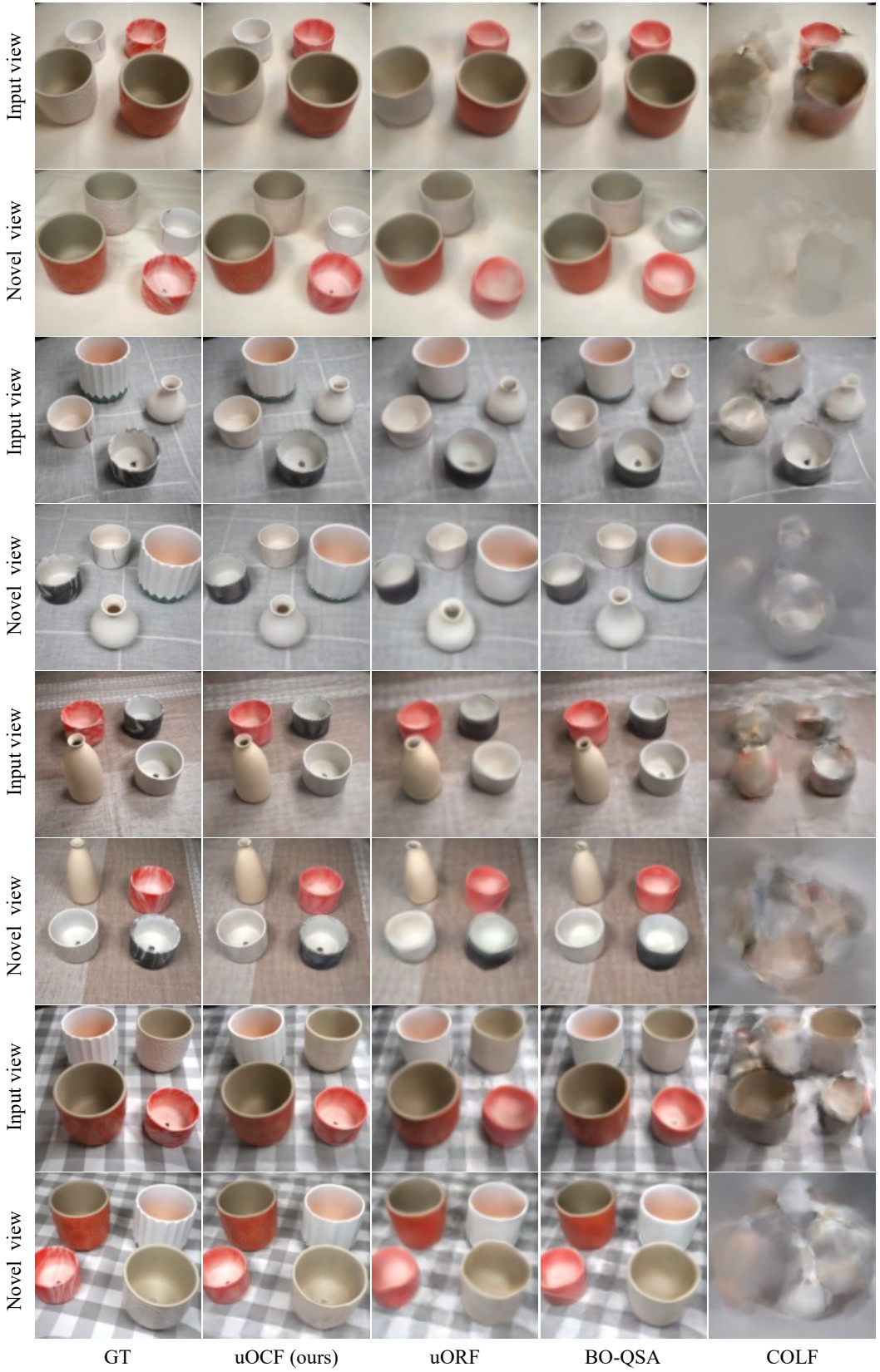

Figure 17: Qualitative comparison results on the Planters dataset.

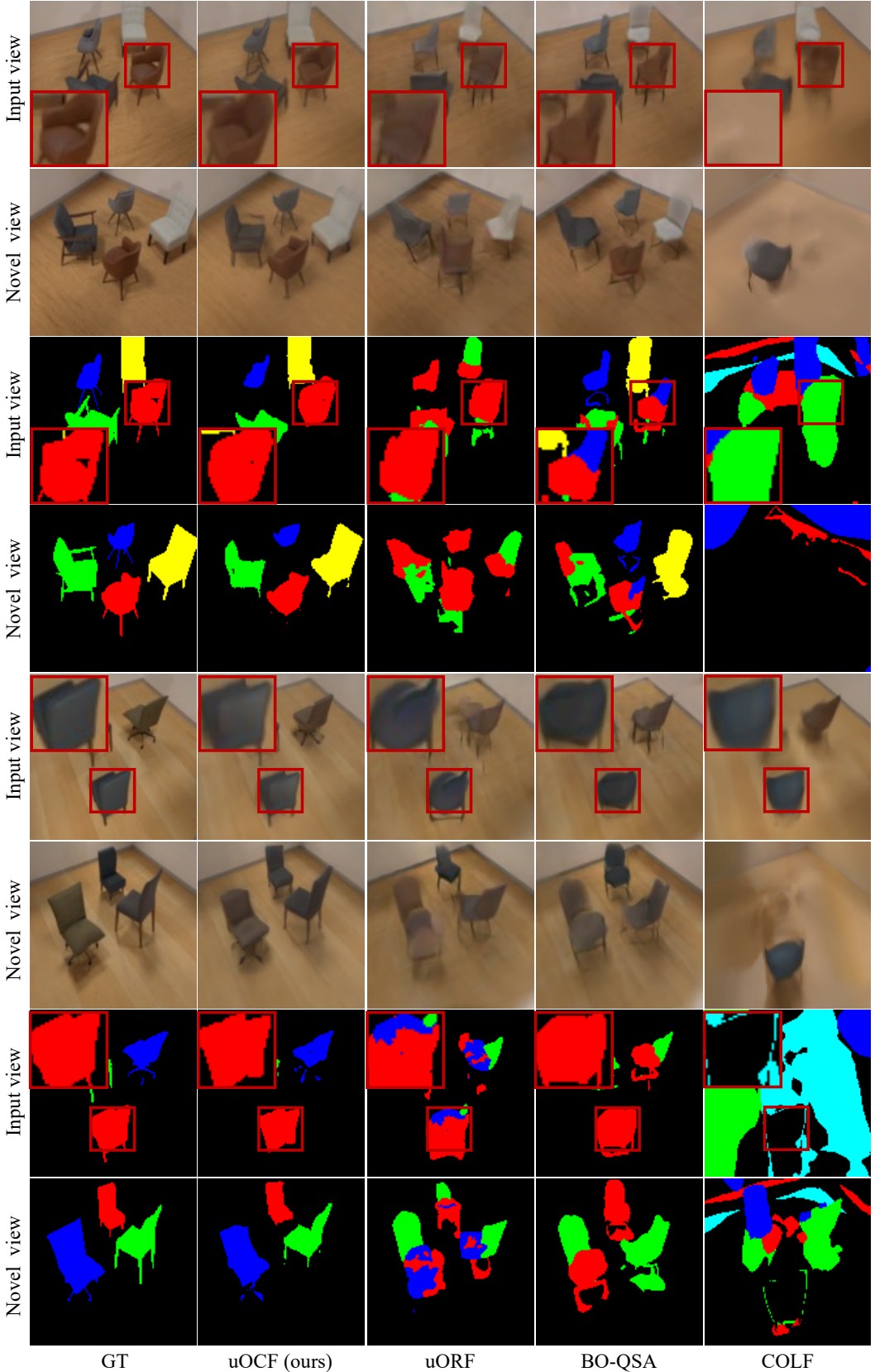

Figure 18: Additional segmentation and view synthesis results on the Room-Texture dataset.

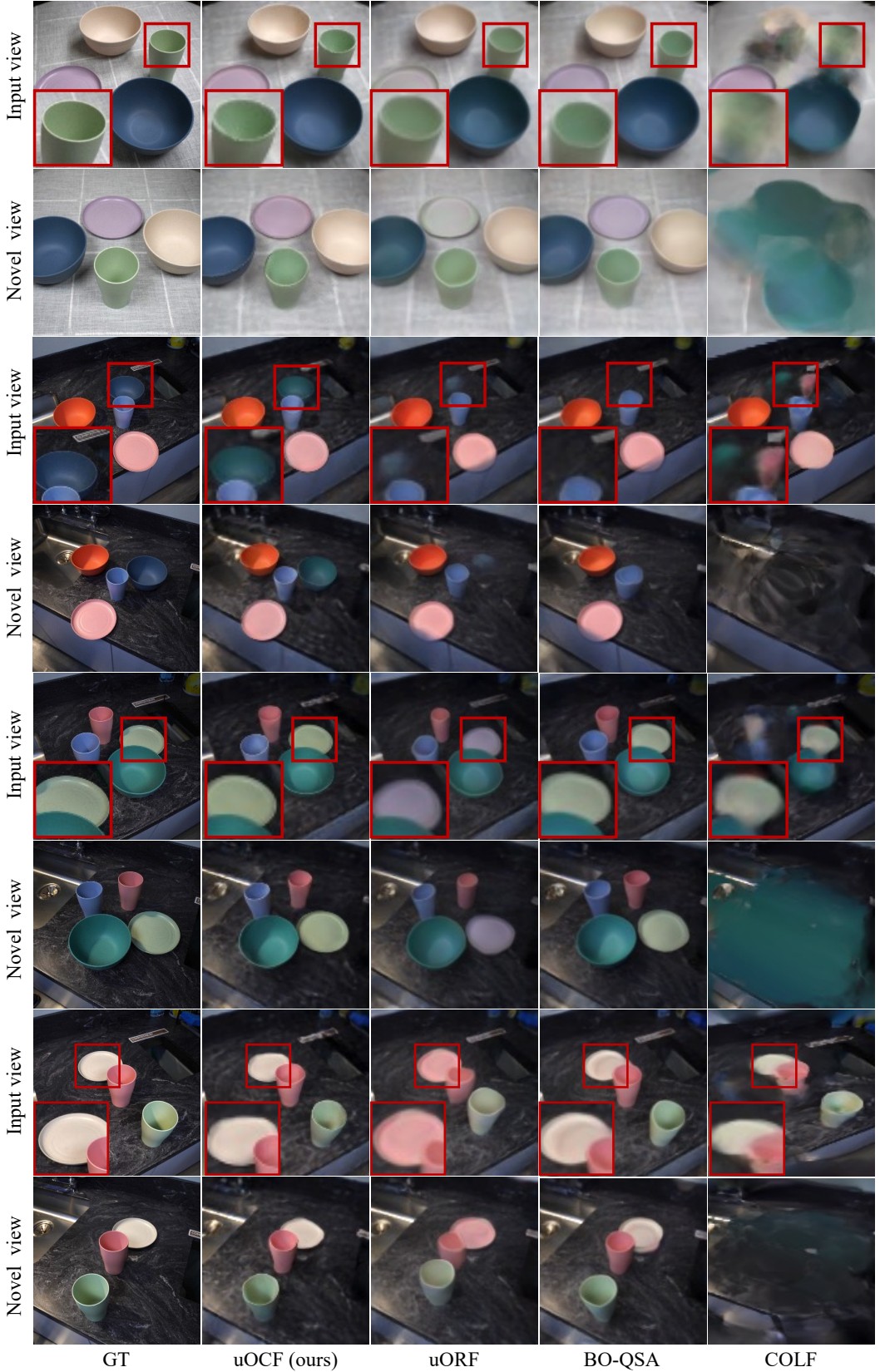

Figure 19: Additional view synthesis results on the Kitchen-Matte dataset.

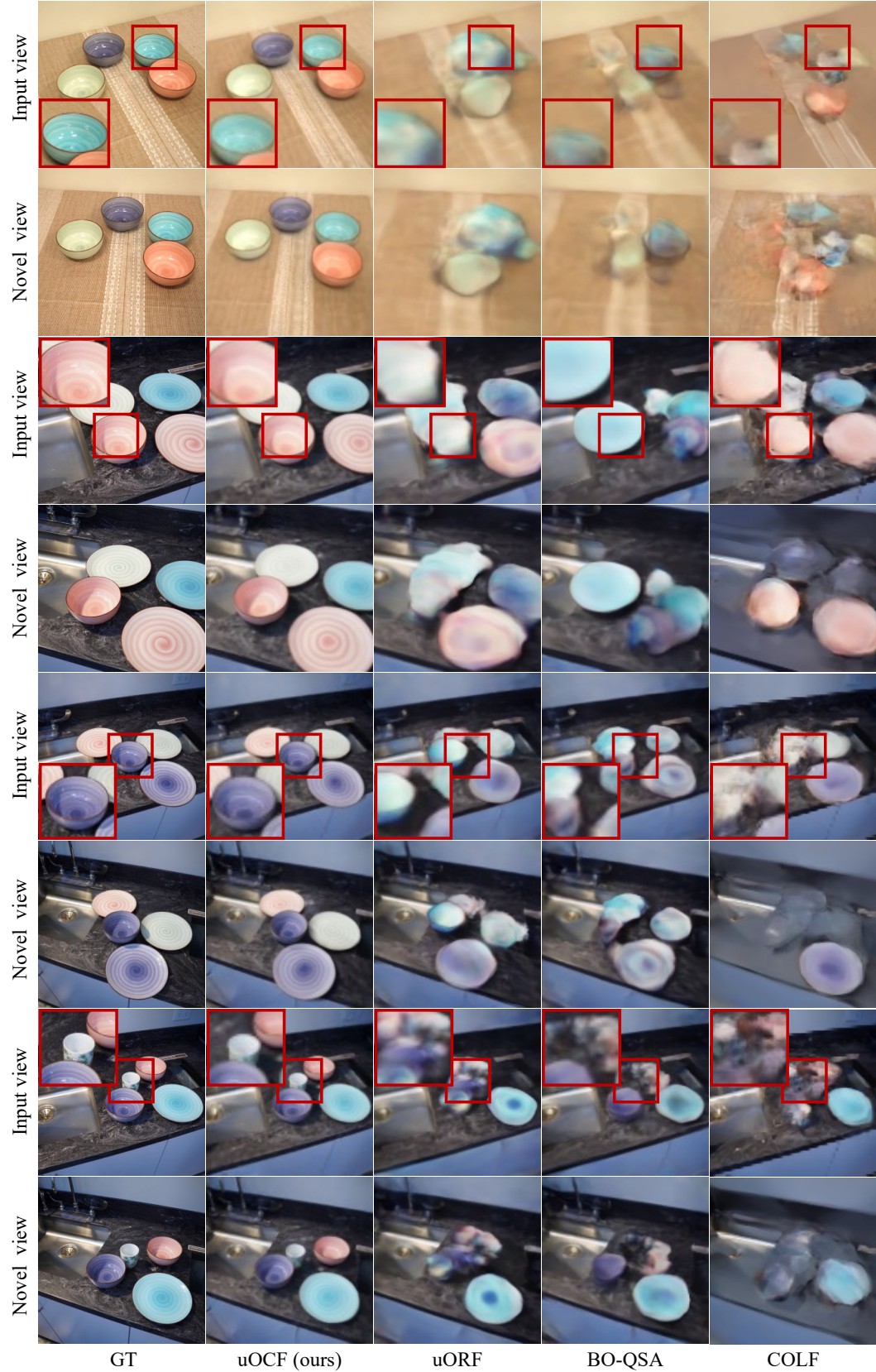

Figure 20: Additional view synthesis results on the Kitchen-Shiny dataset.

