# OpenReview forum: "Unsupervised Discovery of Object-Centric Neural Fields"
_TMLR — Accepted by TMLR_

### Review · Reviewer_q61i · 2024-11-08

**Summary Of Contributions:**

The paper proposes an algorithm for unsupervised 3D object discovery from a single image (while being trained with more views), unsupervised Object-centric Neural Fields (uOCF), where the object representation is translation-invariant. This is achieved by disentangling the object’s latent representation (of its shape and appearance) from its 3D location, which is treated as an additional attribute. This enhancement enables the model to generalize to unseen spatial configurations (e.g., different placements of objects). The model is evaluated on real and synthetic scenes, and the results demonstrate improved visual reconstructions, generalization capabilities and better sample efficiency than prior methods.

**Audience:**

Yes

**Claims And Evidence:**

Yes

**Requested Changes:**

* Page 5: you mention the process of “unprojection”. I believe this requires more explanation. E.g., you mention the utilization of a monocular depth estimation, where did this come from?
* Test-time adaptation: you mention “It requires only a minimal single-image test-time optimization to adapt from one synthetic dataset to real-world images with unseen objects.” and “a fast test-time optimization using a photometric reconstruction loss on the input view only.”. I couldn’t find any details about that, could you please explain the process here?
* Hyper-parameters: the loss weights (reported in the appendix) seem oddly specific–is there intuition or just a sweep to find optimal values?
* Limitations: I appreciate the current discussion on limitations. I would appreciate an additional discussion (can be in the appendix) on the training stability, as slot-attention based methods are prone to training instability (e.g., with the same hyper-parameters, different runs might fail in separating objects to slots). How consistent are the results between different runs with the same hyper-parameters?
* See “Weaknesses” and “Minor”.

**Strengths And Weaknesses:**

**Strengths**:
* Improved reconstruction quality. Position/coordinates matter.
* Improved sample efficiency (training with 10 scenes produces reasonable results).
* Evaluated on real datasets.
* Ability to edit 3D scenes.
* The method generalizes (spatially) with little-to-no post-training effort.
* (A promise of) open-source code.


**Weaknesses**:
* The method requires pre-training on synthetic single-object datasets before it can be trained for multi-object scenes. I think that compared to other methods, this is a limitation. In addition, there are several heuristics employed that are required for the reported performance (e.g., “dropping distant samples from the predicted object positions after a few training epochs“, “we incorporate the depth ranking loss with pre-trained monocular depth estimators and background occlusion regularization to minimize common floating artifacts in few-shot”). It is a weakness because it makes the method more complex and more complicated to train; however, making things work is also an important feat in this field, so I wouldn’t say it is a significant weakness.
* Small number of objects (trained on $K=4$ and tested with $K$ up to $10$). What are the considerations required for scaling up for more objects?
* It seems that many recent methods for unsupervised object discovery are based on the (slot-) attention mechanism (and I appreciate the comparison with slot-attention in the appendix); however, the idea of separating to background and foreground, and the object latent to explicit attributes, such as position and appearance, is not new and dates back to patch/glimpse-based object-centric approaches (e.g., SPACE and SCALOR which you cited), but also apparent in recent methods ([4], [5]). It seems that while the community is converging to the attention-based approach, the ideas from the other object-centric families (patch/glimpse, particles) still prove to be crucial. This paper adopts the position attribute (and maybe future work will discover that other attributes, such as depth or scale, can also help in generalization), and a more transparent discussion on the comparison for the 2D object discovery literature would help highlight the contribution of this paper.

**Minor**:
* Equation 1: what exactly is $D^s$ (specifically, what is $s$)?
* Equation 4: can you explain what $E^{\text{abs}}$ is? To my understanding, it is the absolute normalized coordinates of all the $N$ patches. Also, in Eq. 5 (the matrix), you say that $h_1$ takes in variables of dimension 4, but to my understanding  $E^{\text{abs}}$ is of dimension 2.
* Related work:
  * Since you are using DINO as a pre-trained feature extractor for object discovery, I believe a citation of DINOSAUR [1] is due.
  * Missing (early) related work (not asking to directly compare with): [2], [3]
  * Missing related work for “unsupervised object discovery”: [4], [5], [6], [7]
  * In general, a recent survey [8] of object-centric representations has categorized the different families (not expecting to cite all the works from there of course).



[1] Seitzer, Maximilian, et al. "Bridging the Gap to Real-World Object-Centric Learning." The Eleventh International Conference on Learning Representations. (https://arxiv.org/abs/2209.14860)

[2] Wang, Tianyu, Miaomiao Liu, and Kee Siong Ng. "Spatially invariant unsupervised 3D object-centric learning and scene decomposition." European Conference on Computer Vision. Cham: Springer Nature Switzerland, 2022. (https://arxiv.org/abs/2106.05607)

[3] Henderson, Paul, and Christoph H. Lampert. "Unsupervised object-centric video generation and decomposition in 3D." Advances in Neural Information Processing Systems 33 (2020): 3106-3117. (https://arxiv.org/abs/2007.06705)

[4] Daniel, Tal, and Aviv Tamar. "Unsupervised Image Representation Learning with Deep Latent Particles." International Conference on Machine Learning. PMLR, 2022.  (https://arxiv.org/abs/2205.15821)

[5] Daniel, Tal, and Aviv Tamar. "DDLP: Unsupervised Object-centric Video Prediction with Deep Dynamic Latent Particles." Transactions on Machine Learning Research. (https://arxiv.org/abs/2306.05957)

[6] Löwe, Sindy, et al. "Complex-Valued Autoencoders for Object Discovery." Transactions on Machine Learning Research. (https://arxiv.org/abs/2204.02075)

[7] Gopalakrishnan, Anand, et al. "Recurrent Complex-Weighted Autoencoders for Unsupervised Object Discovery." arXiv preprint arXiv:2405.17283 (2024). (https://arxiv.org/abs/2405.17283v3)

[8] Villa-Vásquez, José-Fabian, and Marco Pedersoli. "Unsupervised Object Discovery: A Comprehensive Survey and Unified Taxonomy." arXiv preprint arXiv:2411.00868 (2024). (https://arxiv.org/abs/2411.00868)

---

> ### Author Response · Authors · 2024-11-28
> **Response (1/2)**
>
> ## Response to questions
>
> We appreciate the reviewer’s thoughtful feedback and the opportunity to clarify and expand on several aspects of our work. Below, we address the comments in detail.
>
> **Q1. Pre-training on Synthetic Datasets and Heuristics**
>
> We acknowledge the concern regarding the additional complexity introduced by pre-training on synthetic single-object datasets. This step is crucial to ensuring robust generalization to real-world scenarios by enabling the model to learn object priors from large-scale datasets like Objaverse-LVIS. These priors are then leveraged to handle diverse environments, including Room-Texture, Room-Furniture, Kitchen-Matte, and Kitchen-Shiny datasets.
>
> Regarding the heuristics employed, such as “dropping distant samples” and incorporating the depth ranking loss, we emphasize that these components contribute to the performance, but they are not our major contribution. As demonstrated in the ablation studies (Table 6), our method retains significant performance advantages over baselines even without these additional losses, underscoring the core effectiveness of our object-centric prior learning approach.
>
>
> **Q2. Scalability for a Larger Number of Objects**
>
> We appreciate your interest in scalability. Currently, memory usage is the primary bottleneck, as each slot queries the NeRF network for num_rays $\times$ num_samples_per_ray, resulting in linear memory scaling with the number of slots. Object-centric sampling partially mitigates this issue, but training on K=4 objects at a resolution of $64 \times 64$ already requires approximately 40GB of memory.
>
> To address scalability, we plan to explore more memory-efficient NeRF implementations and other 3D representations, which could enable training with a larger number of objects. Additionally, we acknowledge the potential impact of scaling on model performance and training time, which we will investigate in future work.
>
>
> **Q3. Relation to Previous 2D Object-Centric Learning Approaches**
>
> We appreciate the reviewer highlighting the connection between our work and prior object-centric learning methods and agree that separating background and foreground, as well as using object attributes such as position, has been explored in 2D object discovery literature. However, adapting these ideas to 3D presents unique challenges and has not been explored by previous works.
>
> Unlike 2D images, where coordinate frames are well-defined, 3D representations require careful consideration of coordinate systems. In existing methods, representing objects in viewer or world coordinates can lead to significant variations in object representations due to subtle changes in object location or camera movements. To address this limitation, our method introduces an object-centric translation-invariant representation. Each object is represented in a local coordinate frame centered on itself, ensuring generalization to unseen object locations during inference.
>
> Furthermore, we emphasize that 3D object discovery requires scene-level understanding, making it fundamentally different from 2D approaches. To provide greater transparency, we will expand the discussion of related work on 2D object-centric learning in the revised manuscript and highlight how our method builds upon and differentiates itself from these foundational approaches.
>
> **Q4. Clarification on Eq. (1)**
>
> We apologize for the typo in the manuscript. The term $D_s$ in Equation (1) is equivalent to $D$, which represents the latent dimension of the model.
>
> **Q5. Clarification on Eq. (4)**
>
> $E^{abs}$ refers to the normalized 2D grid defined as follows:
>
> $$\mathop{flatten}\left(
> \begin{bmatrix}
> (-1+\frac{1}{W}, -1+\frac{1}{H}) & (-1+\frac{3}{W}, -1+\frac{1}{H}) & \cdots & (1-\frac{3}{W}, -1+\frac{1}{H}) & (1-\frac{1}{W}, -1+\frac{1}{H}) \\\\ (-1+\frac{1}{W}, -1+\frac{3}{H}) & (-1+\frac{3}{W}, -1+\frac{3}{H}) & \cdots & (1-\frac{3}{W}, -1+\frac{3}{H}) & (1-\frac{1}{W}, -1+\frac{3}{H}) \\\\
> \vdots & \vdots & \ddots & \vdots & \vdots \\\\ (-1+\frac{1}{W}, 1-\frac{3}{H}) & (-1+\frac{3}{W}, 1-\frac{3}{H}) & \cdots & (1-\frac{3}{W}, 1-\frac{3}{H}) & (1-\frac{1}{W}, 1-\frac{3}{H}) \\\\
> (-1+\frac{1}{W}, 1-\frac{1}{H}) & (-1+\frac{3}{W}, 1-\frac{1}{H}) & \cdots & (1-\frac{3}{W}, 1-\frac{1}{H}) & (1-\frac{1}{W}, 1-\frac{1}{H}) \\\\
> \end{bmatrix}\right)\in \mathbb{R}^{N\times2}
> $$
>
> Subsequently, $\mathbf{E}^{\mathrm{pos}}_i := \mathrm{concat}([\mathbf{E}^{\mathrm{abs}}-\mathbf{p}_i^{\mathrm{img}}, \mathbf{p}_i^{\mathrm{img}}-\mathbf{E}^{\mathrm{abs}}])$ concatenates two matrices of size $N \times 2$ to form a matrix of size $N \times 4$. We also correct the typo regarding background positional encoding and clarify that $h_1$ maps $\mathbb{R}^4$ to $\mathbb{R}^N$. Thank you for catching these errors.

---

> ### Author Response · Authors · 2024-11-28
> **Response (2/2)**
>
> ## Response to requested changes
>
> **Q6. References**
>
> We have updated the manuscript to include all these additional references and a pointer to the recent survey paper [1]. These citations provide a broader context and connect our work to existing literature in object-centric learning.
>
> [1] Villa et al. *Unsupervised Object Discovery: A Comprehensive Survey and Unified Taxonomy*. arXiv, 2024.
>
> **Q7. The Unprojection Process**
>
> As described in Appendix C.3, we utilize a monocular depth estimator [2] to predict per-pixel depth for input images. Given the inherent scale ambiguity of monocular depth predictions, we estimate a global scaling term following the procedure in Sec. 3.2. This ensures consistency across scenes and aligns the depth predictions with the 3D object-centric framework. We have added this information to Sec. 3.2 to make it clearer.
>
> [2] Ranftl et al. *Towards Robust Monocular Depth Estimation: Mixing Datasets for Zero-Shot Cross-Dataset Transfer*. TPAMI, 2022
>
>
>
> **Q8. Details on test-time optimization**
>
> For test-time adaptation, we fine-tune the model on the input view with our proposed loss function (Eq. 7) for 1,000 iterations at a resolution of $128 \times 128$. We employ the Adam optimizer with a $1 \times 10^{-4}$ learning rate. This optimization process takes approximately 3 minutes on a single A6000 GPU. These discussions have been added to Appendix C.3.
>
> **Q9. Clarification on hyperparameter choice**
>
> Due to the high computational cost of training (approximately 1.5 days for Stage 1 and 4.5 days for Stage 2), we were unable to perform an exhaustive hyperparameter sweep. The weights for the reconstruction and perceptual losses are inherited from uOCF [1], while other weights were chosen to ensure comparable magnitudes across all losses. These discussions have been added to Appendix C.3.
>
> **Q10. Discussion on Training Stability**
>
> We acknowledge training instability, particularly in the early stages, where models may collapse even with identical hyperparameters. In practice, we terminate experiments showing early collapse and repeat trials until stable training is observed. Approximately 50% of experiments succeed under these conditions. For baselines failing on complex datasets, we ensure consistent results by conducting at least five independent trials. We plan to investigate strategies to mitigate instability, such as gradient clipping or alternative initialization methods, in future work. These discussions have been added to Appendix E.

---

> > ### Comment · Reviewer_q61i · 2024-12-05
> > **Thank you**
> >
> > I thank the authors for the clarifications and additional details which resolved my concerns, I appreciate the effort in performing more experiments and refining the manuscript.

---

### Review · Reviewer_agc3 · 2024-11-19

**Summary Of Contributions:**

The paper proposed a method to predict 3D object centeric neural field using single RGB image input. The framework presented in this paper aims to create Nerf for each object instead of previous methods' approach where a single Nerf is used for the whole scene. To generalize to the real-world scenes, the paper focuses on disentangling the learning of object intrinsics and the extrinsic separately. The proposed approach significantly improves systematic generalization, enabling unsupervised learning of high-fidelity object-centric scene representations from sparse real-world images. The approach allows for the discovery of visually rich objects from a single real image, allowing for applications such as 3D object segmentation and scene manipulation.

**Audience:**

Yes

**Claims And Evidence:**

Yes

**Requested Changes:**

1. The experiments and results are limited to two set of environments, would like to see more diverse datasets
2. Lack of ablation experiments to show how each part of the model contribute to the overall performance

**Strengths And Weaknesses:**

Strengths:
1.The paper studies a critical problem in object-centric learning without any human annotations, especially discovering objects in 3D space.
2.Using posed multi-views as supervision signals to learn Nerf is interesting to provide more information and constraints for object discovery.

Weakness:
1. The experiments and results are limited to two set of environments, would like to see more diverse datasets
2. Lack of ablation experiments to show how each part of the model contribute to the overall performance

---

> ### Author Response · Authors · 2024-11-28
> **Response**
>
> Thank you for your time and feedback! We respond to each of your comments below, and we summarize all changes in the general response.
>
> **Q1. Experiments on Additional Datasets.**
>
> We thank the reviewer for their suggestion regarding more diverse datasets. Our method has been evaluated on four datasets: two synthetic datasets and two real-world datasets. The synthetic datasets feature simple indoor backgrounds with objects from a wide range of categories, while the real-world datasets focus on kitchen scenes with varied textures and lighting conditions. Compared to prior work, which were usually test exclusively on synthetic scenes, our evaluation settings already incorporate greater complexity and diversity.
>
> To address the reviewer’s concern, we have introduced an additional real-world dataset, “Planters,” which includes plant pots and vases arranged on a table covered with tablecloths. We believe this expands the diversity of our evaluation settings and provides further evidence of our method’s robustness. We have included additional quantitative and qualitative results for this dataset in Appendix D.
>
>
> **Q2. Ablation Studies on Model Components.**
>
> We appreciate the reviewer’s emphasis on the importance of ablation studies. We would like to highlight that Section 4.3 (Tables 5 and 6) of the manuscript provides a detailed analysis of the contributions of key components of our method. Specifically, we evaluate:
> - Translation-Invariant Design: Demonstrating its critical role in enabling generalization to unseen object locations.
> - Object Prior Learning: Highlighting how pre-training on object-centric datasets improves downstream performance.
> - Technical Improvements: Including the use of DINO ViT, standard attention mechanisms, depth and occlusion losses, and the object-centric sampling strategy.
>
> Each component is individually evaluated, with quantitative results that highlight its impact on the model’s overall performance. These studies aim to provide a comprehensive understanding of how our design choices contribute to the success of our approach. However, if there are specific components the reviewer feels are missing from our analysis, we would be happy to address them in the revised manuscript.

---

### Review · Reviewer_L2jR · 2024-11-20

**Summary Of Contributions:**

The work proposes Unsupervised Discovery of Object-Centric Neural Fields (uOCF), a method for explicitly identifying and representing 3D objects within a scene via learned 3D real world coordinates and a latent representation. Conditioning on each object's latent representation, a set of neural radiance field decoders learned per object (and additionally one for background) can be queried continuously in 3D space and consequently rendered to novel 2D views.

The contributions are 3 fold:

- The proposed architecture which addresses the issue of previous implicit object representations that did  not disentangle object position from object representation. The architecture in this work learns the two separately, allowing learned representations to be 3D position agnostic.
- A method for improving generalization is provided by learning object priors from synthetic data. Specifically, the authors propose a series of training steps: first pre-training on vast number of simple easy to generate scenes of single objects from the ObjectVerse dataset, and then subsequently training on more complex scenes of synthetic or real data, enabling the model to develop an understanding of object representations before encountering a further unseen scene.
- 4 datasets are provided: 2 synthetic (one consistent of armchair items with varying backgrounds and one of bedroom items), as well as 2 real datasets (one consistent of kitchen table top views with matte objects, and the other with shiny objects)

uOCF is benchmarked against 3 baseline methods (uORF, BO-QSA, and COLF) on 3 demonstrative tasks of unsupervised object discovery in 3D (i.e. successful segmentation of items), novel view synthesis (visual reconstruction from unseen viewpoints), and 3D scene manipulation (object translation and removal). uOCF outperforms all baselines on these benchmark tasks.

Additionally the work explores the generalization capacity of uOCF via an evaluation of sample efficiency (training from as little as 10 scenes, iteratively up to the full dataset size, and demonstrating varying degrees of object recognition and novel view rendering capacity). The work also demonstrates the ability of uOCF to perform zero-shot adaptation to unseen objects (real world captures) leveraging only a single view from 2 new scenes.

Ablation studies are also conducted to confirm the relative contributions of the translation invariance and object prior pre-training aspects of the proposed method. The authors additionally confirm that the model is robust to the number of queried objects, and an excess of objects can be queried without leading to over segmentation.

**Audience:**

Yes

**Claims And Evidence:**

Yes

**Requested Changes:**

- Include more explicit description of the training phases for uOCF (this is already briefly noted at the end of Object Prior Learning, but would be more clear if e.g. given a small thorough section within Model Training including such additional details such as, when test time optimization is used - such as for zero-shot case - and when it is not.)

- Describe BO-QSA and COLF more explicitly within related works, and briefly explain why the 3 baselines were chosen for benchmarking against uOCF.

- Should have a brief note on what exact data each of the 3 baselines are able to pre-train / train on & are given access to (for example, is object prior learning applied for uORF?) Looking at the uORF method, it is plausible that the full uOCF training scheme of object prior learning and subsequent training / evaluation on the given split, could be equally applied for training the uORF baseline as well. Making explicit in the paper, what exact data uORF had access to in this work, would greatly support ease of understanding.

- For the claim of zero-shot demonstration on real world scenes to be kept - the experiment should be performed on a larger number of real world scenes than 2. Perhaps 3-5 scenes at least, to make a qualitative claim.

- Baselines should ideally be fairly benchmarked on the zero-shot task too, in order to demonstrate relative performance.


Minor / Formatting:
- Add space to "Texturescenes" on page 9

**Strengths And Weaknesses:**

Strengths:

Architecture - The proposal of an architecture explicitly addressing translation invariance is compelling, as a method for enhancing generalization across various scene configurations and layouts. The work also provides a clear ablation study, demonstrating the relative utility of the translation invariance aspect of the model.

Object prior learning - The object prior learning schedule is an interesting proposal, demonstrating a successful approach to leveraging simple synthetic objects to better condition a model to perform on real data. The successful synthetic to real transfer is a nice example of how such easy to collect synthetic data can be leveraged. Zero-shot demonstration on unseen real world scenes is particularly compelling due to this reason.

Weaknesses:

Lack of clarity around what data the baselines are able to have access to train on (e.g. is object prior learning applied for uORF?) For each of the 3 baselines, it would need to be very explicit - how does the training data provided compare to that for uOCF.

Large amount of pre-training - several training stages required (pre-training on single objects, training on the train set of scenes, training/optimization for purpose of discovery on unseen scene in the case of zero-shot.)

Claim on zero-shot is not sufficiently supported - a primary practical application is in the zero-shot case - as for most scenes, there will not be 1000s of related scene configurations to learn from. However, the zero-shot case is only demonstrated for 2 scenes. A larger number of scenes, with a concrete diversity of real world items, would be more appropriate to support the claim on zero-shot.

---

> ### Author Response · Authors · 2024-11-28
> **Response**
>
> Thank you for your constructive feedback! We respond to each of your questions below, and we summarize all changes in the general response.
>
> **Q1. Clarification on Baseline Choices.**
>
> We chose uORF, BO-QSA, and COLF as baselines because they represent state-of-the-art methods for object-centric representation learning. Following your suggestion, we have expanded Section 2 to include a more detailed discussion of these approaches and their relevance to our method.
>
> **Q2. Details on Baseline Implementation**
>
> For all baselines, we followed the original implementations provided by the authors and trained them on multi-object scenes without incorporating object prior learning. To clarify, these methods did not utilize pre-trained weights or synthetic single-object datasets as part of their training pipeline. We have added a concise description of baseline implementation details in the revised manuscript to ensure transparency and reproducibility.
>
> **Q3. Baselines with Object Prior Learning.**
>
> To ensure fair comparisons, we conducted additional experiments where object prior learning was incorporated into the baseline methods. Below are the results:
>
> | Method                           | LPIPS ↓ | SSIM ↑ | PSNR ↑ |
> |----------------------------------|---------|--------|--------|
> | uORF                             | 0.336   | 0.602  | 19.23  |
> | uORF + object prior learning     | 0.193   | 0.714  | 22.78  |
> | BO-QSA                           | 0.318   | 0.639  | 19.78  |
> | BO-QSA + object prior learning   | 0.129   | 0.766  | 24.00  |
> | COLF                             | 0.397   | 0.561  | 18.30  |
> | COLF + object prior learning     | 0.290   | 0.709  | 21.66  |
> | uOCF (ours)                      | 0.049   | 0.862  | 28.58  |
>
> These results demonstrate that even with the incorporation of object prior learning, uOCF significantly outperforms existing methods due to its translation-invariant object representation, which enhances generalization and data efficiency. We have added these experiments to Appendix D.
>
>
> **Q4. Large amount of pre-training - several training stages required (pre-training on single objects, training on the train set of scenes, training/optimization for purpose of discovery on unseen scene in the case of zero-shot.).**
>
> We appreciate the concern regarding the multiple training stages in our pipeline. It is important to note that the pre-training stage on single-object scenes is not dataset-specific; the same synthetic dataset is reused across all experiments, making it highly scalable and efficient to generate. Furthermore, the pre-training step is designed to ensure robust generalization, particularly for downstream tasks requiring adaptation to unseen spatial configurations.
>
>
> **Q5. Clarification on uOCF's training procedure.**
>
> To learn object priors, we generate a synthetic dataset of over 8,000 scenes. Each scene contains one object, sampled from a high-quality subset of Objaverse-LVIS, placed against a room background. These objects span over 100 categories. The synthetic dataset is easy to generate and scalable, making it ideal for learning object priors for all our experiments.
>
> In the second stage, the number of foreground object queries is set to $K=4$. We initialize the model with the pre-trained weights from the object prior learning stage and train it on multi-object scenes. Once trained, our model can perform direct inference on images with spatial configurations that differ from those seen during training. Additionally, our model can adapt to unseen environments through efficient test-time optimization.
>
> Following your suggestion, we have added these discussion to the begining of Sec. 4.
>
>
> **Q6. Additional zero-shot evaluations**
>
> To strengthen our zero-shot claim, we evaluated our method on two additional real-world, phone-captured scenes, expanding our zero-shot evaluations to four real-world scenes. In addition, unlike in Figure 9 where we follow the original implementation of baseline methods, we incorporated object prior learning into their pipelines for a fair comparison. We observed that while object prior learning improved baseline performance (as discussed in Q3), they still fail on the zero-shot generalzation task. Details of these experiments are added to Appendix D.
>
>
> **Q7. Minor/Formatting**
>
> We have corrected the formatting issue. Thank you for pointing this out.

---

### Decision · Action_Editor_ZPzh · 2025-01-21

**Recommendation:** Accept as is

**Comment:**

I am happy to recommend acceptance of the paper as is, given the extensive improvements that authors made (shown in https://openreview.net/forum?id=ScEv13W2f1&noteId=f7YvkZ6Kia) to address the comments of the reviewers.

**Audience:**

This paper will be of interest to computer vision, machine learning, and even robotics researchers.

**Claims And Evidence:**

The paper examines the question of whether translation invariance is a helpful requirement for unsupervised object discovery. The paper shows that the proposed model (uOCF) that leverages translation invariance, by disentangling the object's location from
its latent representation, has better sample efficiency and spatial generalization than alternative methods that do not. The paper also relies on another ingredient, an object prior, to show improved results over baselines, while also avoiding confusion about whether improvement is due to translation-invariance vs the object prior via ablation experiments (eg Tables 5 and 6). Therefore the claims of the paper are well supported by evidence.